



# Oxygenated products formed from OH-initiated reactions of
# trimethylbenzene: Autoxidation and accretion
Yuwei Wang[1], Archit Mehra[2], Jordan E. Krechmer[3], Gan Yang[1], Xiaoyu Hu[1], Yiqun Lu[1], Andrew Lambe[3],
Manjula Canagaratna[3], Jianmin Chen[1], Douglas Worsnop[3], Hugh Coe[2], Lin Wang[1,4,5] *
[1] Shanghai Key Laboratory of Atmospheric Particle Pollution and Prevention (LAP[3]), Department of
Environmental Science and Engineering, Jiangwan Campus, Fudan University, Shanghai 200438, China
[2] Centre for Atmospheric Science, School of Earth and Environment Sciences, The University of
Manchester, Manchester, M13 9PL, UK
[3] Center for Aerosol and Cloud Chemistry, Aerodyne Research Inc., Billerica, MA, USA
[4] Collaborative Innovation Center of Climate Change, Nanjing 210023, China
[5] Shanghai Institute of Pollution Control and Ecological Security, Shanghai 200092, China
*Corresponding Author: L.W., email, lin_wang@fudan.edu.cn; phone, +86-21-31243568*
**Abstract.** Gas-phase oxidation pathways and products of anthropogenic volatile organic compounds
(VOCs), mainly aromatics, are the subject of intensive research with attention paid to their contributions to
secondary organic aerosol (SOA) formation and potentially, new particle formation (NPF) in the urban
atmosphere. In this study, a series of OH-initiated oxidation experiments of trimethylbenzene (TMB, $C_9H_{12}$)
including 1,2,4-TMB, 1,3,5-TMB, 1,2,3-TMB, and 1,2,4-(methyl-D3)-TMBs ($C_9H_9D_3$) were investigated
in an oxidation flow reactor (OFR), in the absence and presence of $NO_x$. Products were measured using a
suite of state-of-the-art instruments, i.e., a nitrate-based chemical ionization - atmospheric pressure
interface time-of-flight mass spectrometer (Nitrate CI-APi-TOF), an iodide-adduct chemical ionization -
time-of-flight mass spectrometer (Iodide CI-TOF) equipped with a Filter Inlet for Gases and AEROsols
(FIGAERO), and a Vocus proton-transfer-reaction mass spectrometer (Vocus PTR). A large number of C9
products with 1-11 oxygen atoms and C18 products presumably formed from dimerization of C9 peroxy
radicals were observed, hinting the extensive existence of autoxidation and accretion reaction pathways in
the OH-initiated oxidation reactions of TMBs. Oxidation products of 1,2,4-(methyl-D3)-TMBs with
deuterium atoms in different methyl substituents were then used as a molecular basis to propose potential
autoxidation reaction pathways. Accretion of C9 peroxy radicals is the most significant for aromatics with
meta-substituents and the least for aromatics with ortho-substituents, if the number and size of substituted
groups are identical. The presence of $NO_x$ would suppress the formation of C18 highly oxygenated
molecules (HOMs) and enhance the formation of organonitrates, and even dinitrate organic compounds.
Our results show that the oxidation products of TMB are much more diverse and could be more oxygenated
than the current mechanisms predict.

## 1 Introduction
Oxidation products of volatile organic compounds (VOCs) contribute significantly to the formation of
secondary organic aerosols (SOAs) (Ng et al., 2010; Zhang et al., 2007), which raises a globally ubiquitous
health and environmental concern (Hallquist et al., 2009). There have been numerous studies that aim to
construct detailed VOC oxidation mechanisms to advance our understanding on VOC degradation, SOA
formation, and ozone formation (Atkinson, 1986; Atkinson and Arey, 2003; Atkinson and Carter, 1984;
Kroll and Seinfeld, 2008; Ziemann and Atkinson, 2012). Based on the hypothesis that the products and



kinetics of many unstudied chemical reactions can be proposed by analogy to known reactions of similar
chemical species (Ziemann and Atkinson, 2012) and/or predicted by the structure-activity relationships
(Kwok and Atkinson, 1995), the Master Chemical Mechanism (MCM) is developed as a nearly explicit
chemical mechanism, describing the degradation of numerous VOCs (Bloss et al., 2005; Jenkin et al., 2003;
Saunders et al., 2003). Due to the high complexity of VOC oxidation processes, it is not surprising that
mechanisms leading to the formation of previously unidentified species are still missing.
The formation of highly oxygenated organic molecules (HOMs) through the autoxidation pathway
during VOC oxidation is such an example. HOMs refer to organic compounds typically containing six or
more oxygen atoms that are formed in the gas phase (Bianchi et al., 2019). Autoxidation is a chemical
process where an alkyl peroxy radical ($RO_2$) undergoes an intramolecular hydrogen shift followed by
addition of a molecular oxygen, resulting in a more oxygenated $RO_2$ radical (Crounse et al., 2013; Ehn et
al., 2014). It is an effectively repetitive uni-molecular reaction as the more oxidized $RO_2$ will serve as a
parent $RO_2$ in the next autoxidation reaction, leading to the rapid formation of HOMs in very short time
scales (Bianchi et al., 2019; Jørgensen et al., 2016).
Owing to recent developments in the analytical techniques such as nitrate-anion chemical ionization
mass spectrometry (nitrate CIMS), our knowledge on the autoxidation pathway during the oxidation of
biogenic volatile organic compounds (BVOCs) has been significantly improved. Certain systems, such as
the oxidation of monoterpenes, have been studied extensively, of which ozonolysis has been confirmed as
an important source for HOMs (Ehn et al., 2014; Jokinen et al., 2014). The OH-initiated oxidation is also a
considerable HOM formation source for monoterpenes and isoprene (Krechmer et al., 2015), albeit at lower
yields for monoterpenes containing an endocyclic double bond (Jokinen et al., 2014, 2015; Rissanen et al.,
2015). Detailed mechanisms of monoterpene-derived HOM formation reactions, initiated by ozone or OH,
were investigated through theoretical calculations (Berndt et al., 2016), or by analogy to reactions of similar
chemical species, i.e., cyclohexene (Rissanen et al., 2014). A couple of studies performed H/D isotope
exchange experiments, which can probe the number of hydrogen atoms other than that in C-H, strongly
supporting the proposal of autoxidation mechanisms (Ehn et al., 2014; Rissanen et al., 2014). Research on
other BVOCs, i.e., isoprene and sesquiterpenes (Crounse et al., 2013; Richters et al., 2016; Teng et al.,
2017), and on other oxidants, i.e., $NO_3$ and chlorine (Nah et al., 2016; Wang et al., 2019), indicate the
widespread existence of autoxidation pathways in the oxidation of BVOCs. The products formed from
autoxidation of biogenic precursors have been proven to play a vital role in atmospheric new particle
formation (NPF) because of their low volatility (Ehn et al., 2014; Stolzenburg et al., 2018; Tröstl et al.,
2016).

On the other hand, studies on autoxidation of anthropogenic VOCs are rather sparse. Wang et al. (2017)
theoretically and experimentally showed the autoxidation route of alkylbenzenes to form HOMs in the gas
phase. Identities and yields of HOM products from different aromatics were systematically measured and
the determined molar HOM yields were in the range of 0.1 % to 2.5 %, which are similar to the molar HOM
yields of OH-initiated reactions of BVOCs (Jokinen et al., 2015; Molteni et al., 2018). Currently, aromatics-
derived HOMs are believed to be formed via many reaction pathways, including accretion, bicyclic
intermediate reactions, and multi-generation OH reactions (Berndt et al., 2018b; Garmash et al., 2019;
Zaytsev et al., 2019). The unimolecular isomerization and autoxidation reactions of aromatic peroxy





radicals have been shown to be fast enough to compete with other bimolecular reactions even under NO
concentrations as high as in urban environment (Tsiligiannis et al., 2019).

Trimethylbenzene (TMB) including isomers of 1,3,5-TMB, 1,2,3-TMB, and 1,2,4-TMB is one of the

most common anthropogenic VOCs in urban areas. OH-initiated oxidation of TMB is its dominant chemical
loss in the atmosphere (Atkinson and Arey, 2003), which proceeds either via H atom abstraction from the
methyl substituents or via addition of OH radical onto the aromatic ring (Ziemann and Atkinson, 2012).
The H atom abstraction channel is minor in the OH-induced oxidation reactions of TMB, forming dimethyl-
benzaldehyde. The major channels of OH addition consist of peroxide-bicyclic pathway, phenolic pathway,
and epoxy-oxy pathway (Bloss et al., 2005; Calvert et al., 2002; Jenkin et al., 2003). The three TMB isomers
have different branching ratios for these pathways resulting from the substitution-, site-, and stereo-
specificity, however specific branching ratios are still in debate. Among these pathways, the peroxide-
bicyclic pathway has the highest branching ratio and can form bicyclic peroxy radicals (BPRs), which are
important intermediates that contribute significantly to the formation of HOMs (Wang et al., 2017).
Subsequent reactions of the intermediates will lead to the formation of stabilized products. On the other
hand, the details of the autoxidation mechanisms for anthropogenic precursors remain elusive. Direct
measurements of individual H-shift rates, the detailed structure of HOMs, and a robust quantification of
HOM yields are still lacking. The detailed kinetics for termination reactions of different $RO_2$ are also
ambiguous. Consequently, it is hard to comprehensively judge the TMB oxidation reaction pathways and
products under different atmospheric conditions, and to evaluate the contribution of TMB oxidation to
atmospheric NPF and SOA formation.

In this study, we studied the OH-initiated oxidation of 1,3,5-TMB, 1,2,3-TMB, and 1,2,4-TMB with

a focus on autoxidation and accretion products, via the concurrent usage of a Vocus proton-transfer-reaction
time-of-flight mass spectrometry (Vocus PTR), an iodide-adduct chemical ionization - time-of-flight mass
spectrometer equipped with a Filter Inlet for Gases and AEROsols (FIGAERO Iodide CI-TOF), and a
nitrate-based chemical ionization - atmospheric pressure interface time-of-flight mass spectrometer (Nitrate
CI-APi-TOF). Oxidation of 1,2,4-(methyl-D3)-TMBs was investigated to elucidate the detailed
autoxidation reaction pathway. The influence of $NO_x$ concentration on product distribution was also
investigated.

**112    2 Methods**

As shown in Figure 1, oxidation experiments of TMB were conducted in a Potential Aerosol Mass

(PAM) oxidation flow reactor (OFR, Aerodyne Research, Inc.). A self-prepared VOC cylinder was used to
provide a constant source of gaseous TMB as a reactant. $O_3$/OH was produced in-situ in the PAM and the
relative humidity (RH) was regulated by the PAM setup, which will be introduced in details later. A Vocus
PTR (Krechmer et al., 2018), a FIGAERO Iodide CI-TOF (Lee et al., 2014; Lopez-Hilfiker et al., 2014),
and a Nitrate CI-APi-TOF (Ehn et al., 2014; Eisele and Tanner, 1993) were deployed to detect gaseous
products as well as particulate ones. In addition, an ozone monitor (Model 106-M, 2B technologies) and a
$NO_x$ monitor (Model 42i-TLE; Thermo Fisher Scientific) were utilized to measure trace gas concentrations,
whereas a set of Scanning Mobility Particle Sizer (SMPS, consisting of one TSI Model 3080 Long DMA
and one TSI Model 3776 Condensation Particle Counter) was employed to measure the number size
distribution of submicron aerosol particles.



**OFR.** In this study, the sum of all the flows in the PAM, including a zero air flow, an ozone ($O_3$) flow,
a TMB/$N_2$ flow, and a $N_2O$/$N_2$ flow depending upon experimental conditions, was kept at either 10 or 10.4
slpm (standard litres per minute, standard to 0 ℃, 1 atm), resulting in calculated mean residence times of
approximately 80 seconds. Zero air was generated by a zero gas generator (Sabio Model 1001 Zero Gas
Source). A fraction of the zero air was passed through a Nafion humidifier (Perma Pure Model FC100-80-
6MSS) filled with ultrapure water to achieve the desired RH in the OFR. Ozone was generated by passing
800 sccm (standard cubic centimetre, standard to 0 ℃, 1 atm) of zero air through a separate ozone chamber
and input into the OFR. In order to create a low $HO_2$/$RO_2$ ratio environment to promote the carbonyl and
hydroxyl channels to terminate $RO_2$ radicals, the OFR was operated with only the 254 nm lights on (Lambe
et al., 2019), which is referred to as OFR254 mode in previous studies (Peng et al., 2015). In OFR254 mode,
the primary oxidant production reactions in the OFR are:
$\qquad O_3 + hv(254\ nm) \rightarrow O_2 + O(^1D)$ $\qquad$ $(R1)$
$\qquad O(^1D) + H_2O \rightarrow 2OH$ $\qquad$ $(R2)$
In some experiments, $N_2O$ (99.999%, Air Liquide) was added at the OFR inlet, corresponding to
mixing ratios of 3.4% of the total gas flow rates, which produced $NO_x$ via the following reactions (Lambe
et al., 2017):
$\qquad N_2O + O(^1D) \rightarrow 2NO$ $\qquad$ $(R3)$
$\qquad NO + O_3 \rightarrow NO_2 + O_2$ $\qquad$ $(R4)$
Before each experiment, the PAM OFR was purged with zero air under the OFR254 operation mode
until the signals of acetic acid and other common VOC oxidation products decreased to background levels
of the Vocus PTR and CI-TOF that are described below.
**Vocus PTR.** The newly developed Vocus PTR has a high sensitivity to a wide range of VOCs and
oxygenated volatile organic compounds (OVOCs) (Krechmer et al., 2018; Li et al., 2019; Riva et al., 2019).
Its mass resolving power  (m/Δm = ~12000 at 200 Th, 1 Th = 1 $u/e$, where $e$ is the elementary charge and
$u$ is the atomic mass unit) allows to simultaneously monitor many isobaric species, and even to distinguish
the very minor mass discrepancy (0.001548 $u$) between one deuterium atom and two hydrogen atoms. The
instrument background together with a quantitative calibration by injection of standards was measured
between every two experiments to minimize potential inaccuracies. In our study, the pressure of the
focusing ion-molecule reactor (FIMR) was actively maintained at 1.5 mbar resulting in an E/N of the FIMR
at 110 Td (1 Td = $1 \times 10^{-17}$ V cm$^2$), which was generally a moderate operating condition leading to relatively
little fragmentation of compounds of interest (Gueneron et al., 2015; Yuan et al., 2017).
**FIGAERO-Iodide CI-TOF.** The Iodide-adduct CI-TOF is able to determine elemental compositions
of a suite of atmospheric oxygenated organic species (D'Ambro et al., 2017; Lee et al., 2014; Lopez-Hilfiker
et al., 2016). It has increasing sensitivities toward more polar and acidic VOCs (Lee et al., 2014). The mass
resolution of the Iodide CI-TOF was tuned to be around 3000. The reagent ion ($I^-$) was produced from
permeated $CH_3I$ vapor in $N_2$ by a radioactive source of Am-241 (0.1 mCi). The pressure in the ion-molecule
reactor (IMR) was regulated at 100 mbar, whereas the small segmented quadrupole (SSQ) pressure was set
to be around 2 mbar. The FIGAERO inlet manifold enables the Iodide CI-TOF to measure both gas and
particle compositions at a molecular level (Lopez-Hilfiker et al., 2014). In our study, aerosols were collected
onto a PTFE filter (5μm, Millipore) at 0.96 slpm for 20 min, while the gases were measured simultaneously





via a separate dedicated port. Then, a thermal desorption cycle was started 2 minutes after the FIGAERO
filter was aligned to a heating tube, through which a heated ultra-high purity nitrogen flow was passed and
heated according to a pre-programmed temperature ramp. The ultra-high purity nitrogen was initially held
at 25 °C for 2 min, and then heated at a rate of 10 °C min$^{-1}$ to 200 °C, which was maintained for the
remainder of the temperature ramp (50 min in total).
**Nitrate CI-APi-TOF.** The Nitrate CI-APi-TOF has been increasingly used for the measurement of
low volatility organic compounds (LVOC) and extremely low volatility organic compounds (ELVOCs)
(Ehn et al., 2014; Hyttinen et al., 2015; Jokinen et al., 2014), which mostly have a high O:C ratio. The
resolving power of the Nitrate CI-APi-TOF was up to around 8000 in our study. The selectivity of nitrate
ions keeps the spectrum clean from the more abundant, less oxidized compounds in our experiments. Most
of the detected species were observed exclusively as adducts with $NO_3^-$, a very minor fraction of which
contain odd hydrogen numbers and are hence postulated to be radicals but not presented in this manuscript.
The concurrent use of three mass spectrometers (MSs) with different reagent ions allows us to obtain
a comprehensive picture of the oxidation products of TMB with OH radicals. The detection suitability of
these three instruments for oxidation products with various levels of oxidation has been discussed a lot in
previous studies (Isaacman-VanWertz et al., 2017; Krechmer et al., 2018; Riva et al., 2019). Generally,
Vocus PTR displays selectivity for less oxidized compounds; Iodide CI-TOF favors more oxygenated
species; and Nitrate CI-APi-TOF shows the highest efficiency for the most oxidized compounds. Dimer
products of TMB oxidation are expected to be detected by Nitrate CI-APi-TOF as clusters with $NO_3^-$, which
is due to the potential hydrogen bond donor functional groups in these molecules, inferred from the
abundant oxygen and hydrogen atoms in the formulas. These products should not be detected by Vocus
PTR. One explanation is that these molecules are likely to be fragile and therefore have fragmented owing
to the protonation or the strong electric field in the FIMR of Vocus PTR. Alternatively, these products
might not go through the PEEK tube inlet of Vocus PTR. At the same time, the sample inlet for Iodide CI-
TOF in our experiments is not desirable for the detection of dimer products.
To ensure that the reported signal is truly from the sample flow instead of internal background or
contamination, subtraction of the mass spectra for the OFR background from the samples has been
performed for each instrument. In addition, since this study is mostly concerned with identification of
oxidation products from OH-initiated reactions of TMBs and elucidation of the potential autoxidation
pathway, Nitrate CI-APi-TOF and Iodide CI-TOF were hence not calibrated and only the arbitrary signals
with MS transmission correction (Heinritzi et al., 2016; Krechmer et al., 2018) were compared within the
same instrument. It should then be noted that the relative signal intensities are biased among the MSs
because of their ionization methods and transmission efficiency.
In each experiment, the Vocus PTR was used to confirm the establishment of stable precursor gas
concentrations, and then the pair of 254 nm Hg lamps were turned on to generate the OH radicals and
reaction products were analyzed by the MSs. The input RH in the OFR was kept at a low level and the
voltage of the Hg lamps was slightly tuned in every experiment, so that the OH exposure in the OFR was
close to one oxidation lifetime of TMB (Kurylo and Orkin, 2003), i.e., consumption of 62.3% of the initial
TMB. Under this condition, the production of the first-generation products is generally favored, if the
subsequent loss reactions for these products are assumed to proceed in the same rate.





Table 1 summarizes all the experiments that were performed. Studied were 1,3,5-TMB (≥ 99.0%,
Aladdin), 1,2,3-TMB (Analytical standard, Aladdin), 1,2,4-TMB (≥ 99.5%, Aladdin), 1,2,4-(1-methyl-
D3)-TMB (≥ 95%, Qingdao Tenglong Weibo Technology Co., Ltd., China), 1,2,4-(2-methyl-D3)-TMB (≥
95%, Qingdao Tenglong Weibo Technology Co., Ltd., China), and 1,2,4-(4-methyl-D3)-TMB (≥ 95%,
Qingdao Tenglong Weibo Technology Co., Ltd., China). The structure of these partially deuterated TMBs
can be found in Figure S1. Note that ozone reactions were not taken into account in this study, because
ozone reacts with aromatics at negligible rates, and its reaction rate with oxidation products containing C=C
double bonds is much slower compared with that of OH (Jenkin et al., 1997, 2003; Molteni et al., 2018;
Saunders et al., 2003).
**3 Results and discussion**
**3.1 Characteristics of C9 products**
Figure 2 presents an overview of C7, C8 and C9 products in a carbon oxidation state ($\overline{OS}_C$)-carbon
number ($n_C$) space as observed by three MSs and also those predicted by MCM v3.3.1. Carbon oxidation
state is a quantity that increases with the level of oxidation, which reveals the chemical aging of atmospheric
organics (Kroll et al., 2011). It is evident that more species were detected by the three MSs, and although
there were clear differences between products detected from different MSs, results indicate missing
oxidation pathways in the current versions of the MCM (MCM v3.3.1, available at:
http://mcm.leeds.ac.uk/MCM). Oxygen-containing C9 products were formed by adding functional groups
to the carbon skeleton, whereas C7 and C8 products resulted from carbon-carbon scission of the original
carbon skeleton together with functionalization. A large proportion of C7-C9 products were more oxidized
than those predicted by MCM, hinting the existence of highly efficient oxidation pathways. At the same
time, some of the C7 and C8 products were characterized with unexpected low $\overline{OS}_C$, of which a few were
even less oxidised than the precursor. The observation of these products is another indication for the
existence of missing pathways in the current oxidation mechanisms.
Recent studies have emphasized on the importance of the peroxide-bicyclic pathway in producing
highly oxygenated compounds in the oxidation of alkylbenzenes (Wang et al., 2017; Zaytsev et al., 2019),
which leads to the formation of ring-retaining products. Therefore, here we further investigated C9 products
of TMB oxidation detected by the three MSs (Figure 3). $C_9H_{10}O_{1-6}$, $C_9H_{12}O_{1-7}$, and $C_9H_{14}O_{4-6}$ contributed
to the most of the signal intensities in Vocus PTR (Figure 3a). Compounds with fewer hydrogen atoms than
TMB in Vocus PTR might be formed from hydrogen abstraction reactions. Iodide CI-TOF detected
products with five to seven oxygen atoms (Figures 3b & 3c), which is narrower compared with Vocus PTR
and Nitrate CI-APi-TOF. Molecules with 18 hydrogen atoms were detected only in Iodide CI-TOF, which
is an unexpected high number. These molecules, low in signal intensities in both gas and particle phases,
might be formed from multiple OH attacks since each OH attack can only add two hydrogens in maximum
onto the parent molecule. The species with the highest signal intensities measured in the gas phase appeared
to be $C_9H_{12}O_4$, $C_9H_{12}O_6$, $C_9H_{14}O_5$, and $C_9H_{14}O_6$ in the 1,2,4-TMB + OH experiment, $C_9H_{14}O_5$ and $C_9H_{14}O_6$
in the 1,3,5-TMB + OH experiment, and $C_9H_{12}O_6$ and $C_9H_{12}O_7$ in the 1,2,3-TMB + OH experiment (Figure
3b). Compared with the gas phase, more oxidized particulate products tended to contribute a larger
proportion of signal in FIGAERO-Iodide-CI-APi-TOF (Figure 3c). Nevertheless, the gas phase products





are emphasized in the current study, which can be detected by and compared among the three instruments.
Nitrate CI-APi-TOF detected C9 products containing 12-16 hydrogen atoms and 5-11 oxygen atoms (Figure
3d).
RO$_2$ radicals can react in the absence of NO, to form termination products including carbonyls,
alcohols, and hydroperoxides via the following reactions (Mentel et al., 2015).
$$RO_2 + R'O_2 \rightarrow R_H C = O + R' - OH + O_2 \qquad (R5)$$
$$RO_2 + R'O_2 \rightarrow ROH + R'_H C = O + O_2 \qquad (R6)$$
$$RO_2 + HO_2 \rightarrow ROOH + O_2 \qquad (R7)$$
Here we present a criteria method based on the work of Mentel et al. (2015). For a parent peroxy radical
with a molecular mass of $m$, its termination ought to lead to the formation of a carbonyl, an alcohol, and a
hydroperoxyl, which have a molecular mass of $m$-17, $m$-15, and $m$+1, respectively. Since elemental
formulas as determined by the high-resolution MS do not contain information regarding functional groups
or the structure of a molecule, the identified mass spectral signals could be counted as either one of the
three categories. Listed in Table 2 are detected stabilized oxidation products in categories of carbonyl,
alcohol, and hydroperoxyl, which hints the potential existence of the corresponding peroxy radicals. These
stabilized products all contain six or more oxygen atoms, which meet the definition of HOMs (Bianchi et
al., 2019). $C_9H_{12}O_6$ is the only signal that has been predicted by MCM, assumed to be a hydroperoxyl
product from a ring-opening peroxy radical that goes through multiple OH attack reactions (MCM name:
C7MOCOCO3H), which is unlikely to occur under our experimental conditions. Four pairs of peroxy
radicals, i.e., $C_9H_{13}O_7$• and $C_9H_{13}O_9$•, $C_9H_{13}O_8$• and $C_9H_{13}O_{10}$•, $C_9H_{15}O_7$• and $C_9H_{15}O_9$•, and $C_9H_{15}O_8$• and
$C_9H_{15}O_{10}$•, can be selected from the eight potential peroxy radicals in Table 2. The molecular formulas for
the peroxy radicals within each pair differ by $2 \times O$, which is a first evidence for the autoxidation pathway.
**3.2 Autoxidation mechanisms of 1,2,4-TMB**
The autoxidation pathways were then further elucidated by experiments with isotopically labelled
precursors, 1,2,4-(1-methyl-D3)-TMB, 1,2,4-(2-methyl-D3)-TMB, and 1,2,4-(4-methyl-D3)-TMB, whose
structure is shown in Figure S1.
If an intramolecular hydrogen shift happens during autoxidation with the abstracted hydrogen coming
from a methyl group, molecular oxygen will rapidly attach to this carbon-centred radical to form a new
alkyl peroxy radical (Bianchi et al., 2019 and reference herein). One potential fate of this R-CH$_2$OO• radical
is to lose one of the two remaining hydrogen atoms, forming a carbonyl according to Reaction R5. Thus,
one of the three original hydrogen atoms in the methyl group will leave this molecule after an autoxidation
step (Ehn et al., 2014; Mentel et al., 2015; Molteni et al., 2018; Otkjær et al., 2018; Rissanen et al., 2014;
Wang et al., 2017). In the case of a deuterium abstraction from a methyl-D3 group during the autoxidation,
an oxidation product with two deuterium atoms ($C_xH_yD_2O_z$) will then be formed, which is presumably a
carbonyl. Although an alcohol or a hydroperoxyl could also be formed from a peroxy radical, it is not
suitable to utilize the presence of alcohol and hydroperoxyl products as a criteria to judge the existence of
autoxidation. The hydroxyl channel of deuterated peroxy radicals can lead to the formation of alcohol
products with either 3 or 4 deuterium atoms, depending on the nature of the other reacting RO$_2$. The slow
unimolecular reaction rate of deuterated methyl group corresponds to little formation of the products with





4 deuterium atoms, whereas our MSs cannot differentiate 3 deuterium atoms either from a molecule with
autoxidation and hydroxyl termination or from an untouched methyl-D3 group. On the other hand, the
hydroperoxyl channel would lead to the formation of hydroperoxyl products with 3 deuterium atoms, too.
Therefore, only the carbonyl channel products of a peroxy radical was used to suggest the potential
autoxidation that has occurred.
Table 3 summaries two-deuterium-containing C9 ($C_9H_yD_2O_z$) products that were detected by Vocus
PTR and Nitrate CI-APi-TOF in different isotope labelling experiments: $C_9H_{10}D_2O_6$ in the 1,2,4-(1-methyl-
D3)-TMB + OH experiment by Vocus PTR and Nitrate CI-APi-TOF; $C_9H_{10}D_2O_7$ in the 1,2,4-(1-methyl-
D3)-TMB + OH experiment by Vocus PTR; and $C_9H_{12}D_2O_8$ in the 1,2,4-(4-methyl-D3)-TMB + OH
experiment by Nitrate CI-APi-TOF. $C_9H_{10}O_7D_2$ (234.0703 Th) was expected to be detected by Nitrate CI-
APi-TOF, but unfortunately an undefined peak (located at 295.9827 Th) covered the position where
$C_9H_{10}O_7D_2 \cdot NO_3^-$ (296.0592 Th) was supposed to been identified. $C_9H_{12}D_2O_8$ (252.0814 Th) was not
detected by Vocus PTR, likely owing to either its low proton affinity or its partitioning onto the inlet of
Vocus PTR , given its high O:C ratio and hence low volatility. However, Nitrate CI-APi-TOF was able to
detect this very sticky compound, because the nitrate source is constructed with concentric sample and
sheath flows that minimize the diffusive losses of samples to the source wall. These results indicate that an
intramolecular deuterium-migration happened on the 1-methyl-D3 substituent of the $C_9H_{10}D_3O_4$ and
$C_9H_{10}D_3O_5$ radicals, and the 4-methyl-D3 substituent of the $C_9H_{12}D_3O_7$ radical, respectively, then one
oxygen was added to the resulting alkyl radicals, and the new peroxy radical reacted to form $C_9H_{10}D_2O_6$,
$C_9H_{10}D_2O_7$, and $C_9H_{12}D_2O_9$, respectively.
These three compounds ($C_9H_{10}D_2O_6$, $C_9H_{10}D_2O_7$, and $C_9H_{12}D_2O_9$) did not possess high signal
intensities, because the deuterium transfer reactions are typically significantly slower for D ($^2H$) nuclei than
hydrogen transfer reactions for H ($^1H$) (Bianchi et al., 2019; Wang et al., 2017). There might be other two-
deuterium-containing C9 products in these experiments. However, since many of these signals were at the
instrument detection limits or even lower, the nonideal experimental conditions prevent us from confirming
more such compounds.
Based on the observed signals of two-deuterium-containing C9 products and structures that have been
previously proven to favor H-shift reactions (Otkjær et al., 2018), two plausible formation pathways for the
observed products are proposed.
The first one starts with a BPR of $C_9H_{13}O_5 \cdot$ as shown in Scheme 1, which is the first BPR formed from
$C_9H_{12}$ via the peroxide-bicyclic pathway. The structure of this particular $C_9H_{13}O_5 \cdot$ is different from what is
proposed in MCM v3.3.1, but the position for the initial OH attack, i.e., the 4$^{th}$ carbon on the ring, is feasible
owing to the attraction of a substituted group on its para-position (Li and Wang, 2014), and the subsequent
addition of $O_2$ after the initial OH attack along with bicyclization occurs on the same relative position as
previous studies have suggested (Bloss et al., 2005; Jenkin et al., 2003). The resulting BPR of
$C_9H_{13}O_5 \cdot$ undergoes a hydrogen shift, during which the abstracted hydrogen comes from the methyl
terminal of an allylic group. This hydrogen is much easier to be abstracted, compared to those in a normal
methyl group that are unlikely to go through a hydrogen shift with a peroxy radical (Otkjær et al., 2018).
The new BPR of $C_9H_{13}O_7 \cdot$ then reacts via R5, R6, and R7 to form $C_9H_{12}O_6$, $C_9H_{14}O_6$, and $C_9H_{14}O_7$,
respectively. This pathway is suggested by the observation of $C_9H_{10}D_2O_6$ in the 1,2,4-(1-methyl-D3)-TMB
+ OH experiment. On the other hand, $C_9H_{13}O_5 \cdot$ can alternatively self-react or react with a $HO_2$ radical to



form an alkoxy intermediate, which goes through isomerization and addition of an oxygen to form a BPR
of $C_9H_{13}O_8\cdot$. The stabilized products from $C_9H_{13}O_8\cdot$ include $C_9H_{12}O_7$, $C_9H_{14}O_7$, and $C_9H_{14}O_8$. This pathway
is suggested by the observation of $C_9H_{10}D_2O_7$ in the 1,2,4-(1-methyl-D3)-TMB + OH experiment.
It's noted that in all the three isotope experiments, we also detected products of $C_9H_9D_3O_6$ and
$C_9H_9D_3O_7$ with much higher signal intensities, indicating the existence of other autoxidation pathways.
Thus, it deserves a repeated emphasis here that we only point out feasible pathways that are supported by
our isotope experiments in this work, but do not rule out other possibilities.
The second pathway is described in scheme 2. This pathway starts from a BPR of $C_9H_{13}O_5\cdot$ that is
formed by the initial OH attack and subsequent reactions. MCM v3.3.1 includes a BPR with the same
structure but does not contain the subsequent reactions. The BPR of $C_9H_{13}O_5\cdot$ can be terminated via R5,
forming a stabilized hydroxyl product of $C_9H_{14}O_4$, which is subject to a second OH attack and a following
addition of $O_2$, resulting in a new peroxy radical of $C_9H_{15}O_7\cdot$. There are no systematic investigations on the
effect of a peroxide-bicyclic substitution on the 1,5 H-shift rate constant. However, our data indicate a
hydrogen shift can occur on the 4-methyl group, based on which the structure of $C_9H_{15}O_9\cdot$ is proposed. The
new BPR of $C_9H_{15}O_9\cdot$ is then terminated via R5, R6, and R7, forming stabilized products $C_9H_{14}O_8$, $C_9H_{16}O_8$,
and $C_9H_{16}O_9$, respectively. This pathway is suggested by the observation of $C_9H_{12}D_2O_8$ in the 1,2,4-(4-
methyl-D3)-TMB + OH experiment, though other pathways could result in products with the same formula.

**3.3 Characteristics of C18 HOMs**
Products with 18 carbon atoms were observed in our experiments by Nitrate CI-APi-TOF, all
containing 24-30 hydrogen atoms and 8 or more atoms ($C_{18}H_{24/26/28/30}O_{>8}$) (Figure 4). C18 products with 26
or 28 hydrogen atoms contributed the most of the signal intensities while those generated by 1,3,5-TMB
were the most abundant. Recent studies revealed that long-neglected organic peroxide dimer (ROOR')
formation reactions might be an important source of gas-phase dimer compounds, through which two
peroxy radicals form accretion products consisting of the carbon backbone of both reactants (Berndt et al.,
2018a, 2018b; Zhao et al., 2018).
$$RO_2 + R'O_2 \rightarrow ROOR' + O_2 \qquad\qquad (R8)$$
This reaction has been proved to be another important loss process for $RO_2$ radicals formed via autoxidation.
On account of their extraordinarily low vapor pressure, HOM dimers contribute more significantly to the
formation and growth of atmospheric new particles than HOM monomers.
Our C18 oxidation products have similar ion formulas to the dimer products in recent 1,3,5-TMB
oxidation experiments (Molteni et al, 2018; Tsiligiannis et al., 2019). In our experiments, the formation of
$C_{18}H_{26}O_{8-15}$, $C_{18}H_{28}O_{9-15}$, and $C_{18}H_{30}O_{12-15}$ can be explained by reactions of two $C_9H_{13}O_x\cdot$, one $C_9H_{13}O_x\cdot$ and
one $C_9H_{15}O_x\cdot$, and two $C_9H_{15}O_x\cdot$ respectively. $C_{18}H_{24}O_{8-13}$ with low signal intensities were detected by
Nitrate CI-APi-TOF, hinting that H-abstraction reactions have occurred leading to a lower hydrogen atom
in the product than in the precursor.
Figure 5 summarizes the relative contribution of C9 and C18 products formed from TMB oxidation as
detected by Nitrate CI-APi-TOF. The charge efficiency for C9 and C18 products is assumed to be identical
in Nitrate CI-APi-TOF. Hence, the measured relative abundances of the oxidation products, with
corrections of the transmission function in the MS, can faithfully represent the product distribution in the





experiments. In the Exp. #1-3, the dimers ($C_{18}H_{26}O_{8-15}$) formed from two $C_9H_{13}O_x\cdot$ along those ($C_{18}H_{28}O_{9-15}$) from one $C_9H_{13}O_x\cdot$ and one $C_9H_{15}O_x\cdot$ contributed the most intensity, whereas the most intensive C9
products ($C_9H_{14}O_{5-11}$) could be the alcohol or hydroperoxyl products of $C_9H_{13}O_x\cdot$, or the carbonyl products
of $C_9H_{15}O_x\cdot$ (Table S1). 1,2,3-TMB produced the most C9 products, 1,2,4-TMB the second, and 1,3,5-TMB
the least. An opposite trend was observed for C18 products. Therefore, the reduction of C9 products was
likely due to the dimer formation. Here, we define the C18 fraction as the ratio of the signal intensities of
C18 products to the sum of those of C9 and C18 products in Nitrate CI-APi-TOF, and the C9 fraction in a
similar way. According to our results, the dimer fraction was the highest for aromatics with meta-
substituents and the least for aromatics with ortho-substituents, if the number and size of substituted groups
are identical, while the monomer fraction had an opposite tendency. This can be explained by the
stereoselectivity of accretion formation reactions.
Under our experimental conditions, the C18 dimer fraction in the 1,3,5-TMB experiments was around
86.5%, which is much higher than the dimer fraction of 42.6%-56.5% re-calculated using the measured C9
and C18 signals by Tsiligiannis et al. (2019), 43.3%-52.4% modelled by Tsiligiannis et al. (2019), and 39%
reported by Molteni et al. (2018t). The lack of a m/z-transmission correction in the former two studies could
partially explain the discrepancy (Molteni et al., 2018; Tsiligiannis et al., 2019). On the other hand, this
observation could also be due to the much higher $RO_2$ concentrations in our experiments. The amount of
reacted 1,3,5-TMB in our experiment is around 74.1 ppb ($\sim 1.8 \times 10^{12}$ molecules cm$^{-3}$), whereas in the
experiments of Tsiligiannis et al. (2019) and Molteni et al. (2018), the numbers are 26 ppb ($\sim 6.5 \times 10^{11}$
molecules cm$^{-3}$) and 22.3 ppb ($\sim 5.6 \times 10^{11}$ molecules cm$^{-3}$), respectively.

## 3.4 Influence of NOx

Figure 6 describes the distribution of C9 products detected by Nitrate CI-APi-TOF in the absence of
$NO_x$ (Exp. #1), a low $NO_x$ experiment (Exp. #7), and a higher $NO_x$ experiment (Exp. #8), respectively.
Once $NO_x$ was added, the formation of C9 non-nitrogen products declined down to around 20% of those in
Exp. #1. The production of C9 non-nitrogen products did not decrease much between low $NO_x$ experiment
and higher $NO_x$ experiment, indicating a nonlinear effect of $NO_x$ on the production of C9 non-nitrogen
products. Dinitrates ($C_9H_xN_2O_y$) increased with the $NO_x$ concentration, but C9 organonitrates (ONs,
$C_9H_xNO_y$) slightly reduced in the higher $NO_x$ experiment compared to that in the low one, which indicates
a complex competition between $RO_2 + RO_2$ and $RO_2 + NO_x$.
The observation of C9 products containing 1-2 nitrogen atoms and C18 products with one nitrogen
atom is similar to the results for 1,3,5-TMB oxidation experiments in the presence of $NO_x$ reported by
Tsiligiannis et al. (2019). $NO_x$ can perturb the fate of peroxy radicals by the following reactions (Orlando
and Tyndall, 2012; Rissanen, 2018):
$RO_2 + NO \rightarrow RONO_2$              $(R9)$
$RO_2 + NO_2 \rightarrow RO + NO_2$          $(R10)$
$RO_2 + NO_2 \rightarrow RO_2NO_2$           $(R11)$
Competing with the other $RO_2$ reactions, $NO_x$ can dramatically reduce the formation of C9 non-nitrogen
products. The $NO_x$ levels in the low $NO_x$ experiment (Exp. #7) and higher $NO_x$ experiment (Exp. #8) were





0.8 ppb and 6.5 ppb, respectively. Compared to the ambient values in polluted areas, this $NO_x$/VOC is low.
The $NO_x/(\Delta VOC)$ was around 0.8% in the low NOx experiment and 6.4% in higher $NO_x$ one.
Most organonitrates observed in our study were characterized with 13 hydrogen atoms, as detected by
Nitrate CI-APi-TOF (Figure S2). All of them contained more than 6 oxygen atoms, with molecular formulas
corresponding to bicyclic organonitrates formed from termination reactions of $C_9H_{13}O_x$• with NO or $NO_2$
(i.e., pathway R9 and R11, respectively). The dinitrates were dominated by species with 14 hydrogen atoms
(Figure S3). As suggested by Tsiligiannis et al. (2019), an OH radical could attack a nitrated compound
that is formed from $NO_x$ termination of a peroxy radical, then an oxygen atom is added (similarly to the
reactions from $C_9H_{14}O_7$ to $C_9H_{15}O_7$ in scheme 2), and then the newly formed peroxy radical that have
already contained one nitrogen will be terminated by NO or $NO_2$ again. Therefore, most of the detected
dinitrates were also formed from $C_9H_{13}O_x$•.
Figure 7a describes the relative intensities of C18 HOMs in Exp. #7, and Exp. #8 as detected by Nitrate
CI-APi-TOF, in comparison with their relative intensities in Exp. #1. The relative intensities of most of the
C18 HOMs decreased with the $NO_x/(\Delta VOC)$, while a few of the C18 HOMs including $C_{18}H_{24}O_{13}$, $C_{18}H_{26}O_{13}$,
$C_{18}H_{26}O_{14}$, $C_{18}H_{28}O_{12}$ increased slightly  in the higher $NO_x$ experiment, potentially from a combined effect
of $NO_x$ and OH. The injection of $NO_x$ can compete with the other $RO_2$ reactions, and thus it consumes
peroxy radicals that would otherwise go through accretion reactions, which explains the decrease of most
C18 HOMs. On the other hand, the introduction of $NO_x$ can increase the oxidation capacity in the OFR, as
it does in the ambient environment, leading to the slight enhancement for the few C18 HOMs. After the
addition of $NO_x$, all of the C18 HOMs decreased by more than six times compared with those in no $NO_x$
experiments, indicating that the dimers were more strongly influenced than monomers, which is in
agreement with a previous study (Tsiligiannis et al., 2019).
The C18 ONs with 25 or 27 hydrogen atoms were detected in the $NO_x$ experiments (Figure 7b). Other
C18 products containing nitrogen atoms were not detected. The $C_{18}H_{25}NO_x$ might be formed from reactions
between a $C_9H_{12}NO_x$• radical and a $C_9H_{13}O_x$• radical, or between a $C_9H_{14}NO_x$• and a $C_9H_{11}O_x$• radical, all
of which existed in the system. The $C_{18}H_{27}NO_x$ is most likely to be formed from reactions between a
$C_9H_{14}NO_x$• radical and a $C_9H_{13}O_x$• radical, which were the most abundant C9 radicals. All the C18 ONs
decreased with the increase of $NO_x/(\Delta VOC)$, which is reasonable. Introduction of $NO_x$ into the system
triggered reactions between C9 peroxy radicals and $NO_x$, which consequently reduced the formation of
accretion products like C18 ONs.

## 435    4 Conclusions

The identities and distribution of oxidation products formed from OH-initiated reactions of three
TMBs were obtained with a suite of state-of-the-art chemical ionization mass spectrometers. Our recent
study shows that the ring-retaining products are more oxygenated and quite a lot of carbon-carbon scission
products are missed in the current model, indicating that the degradation products of aromatics are much
more diverse than what is available in MCM (Mehra et al., 2020). Because of its important contribution to
the nucleation and SOA formation in urban areas, the ring-retaining products of TMB deserve a more
detailed characterization. Here we have built on that work by showing the formation pathways of ring-
retaining highly oxygenated products and through identification of accretion products.





With the assistance of three 1,2,4-(methyl-D3)-TMB experiments we have demonstrated that the rapid
formation of HOMs is attributable to the autoxidation pathway during the TMB oxidation. Several plausible
autoxidation pathways for OH-initiated reactions of 1,2,4-TMB were proposed, emphasizing on the ring-
retaining pathways of aromatics, especially the bicyclic-peroxide channel, which is followed by
autoxidation that is not shown in the current models, such as MCM. Oxidation of aromatic VOCs was
shown in our study to produce HOM dimers, which might be underestimated or even completely ignored
in previous studies which utilize techniques not capable of detecting dimers. The structural enhancement
for accretion product formation via the $RO_2 + R'O_2$ reaction has been observed, of which the meta-
substituents was shown to be strongest and ortho-substituents the weakest, though the detailed
stereoselectivity for aromatics remains unclear now.
In the presence of $NO_x$ whose reaction with $RO_2\cdot$ can compete with $RO_2\cdot + RO_2\cdot$ or $RO_2\cdot +$
$HO_2\cdot$ reactions, ONs and dinitrates will be generated via reactions of $NO_x$ with BPRs in 1,2,4-TMB
oxidation system, and dimer products with one nitrogen will be formed via the subsequent reactions. This
is consistent with a recent ambient observations in the polluted environment, where ONs, dinitrates, and
nitrogen-containing dimers presumably formed from BVOCs and alkylbenzenes were detected (Brean et
al., 2019). The formation of ONs and dinitrates from TMB is not linearly depending on the $NO_x$
concentration, which excludes the possibility of extrapolating our laboratory results to ambient conditions.
Nevertheless, the changes of HOM compositions in the presence of $NO_x$, especially the accretion products,
could have an effect on NPF and SOA formation. Previous work has showed that the ring-retaining product
formation at $NO_x$ environment tends to be more important for TMB than other single substituted C9
aromatics, i.e., isopropylbenzene and propylbenzene, which emphasized the significance of TMB ring-
retaining oxidation in the urban environment (Mehra et al., 2020). Further research is needed to acquire a
quantitative understanding of the role of $NO_x$ in HOM formation.
Clearly, these multifunctional gas phase products appear at different stages of the oxidation chain.
These mass spectra can be used as ideal "fingerprints" of TMB oxidation in the ambient gas phase
measurement to elucidate atmospheric oxidation conditions.

*Data availability.* Data related to this article will be available from a persistent repository and upon request
from corresponding authors.

*Supplement.* The supplement related to this article is available online.

*Author contributions.* LW, and YW designed the experiments. YW, GY, XH, and YL carried out the
instrument deployment and operation. AM, JK, and AL provided technical support. YW analyzed the data.
YW, LW, and JK wrote the paper. All co-authors discussed the results and commented on the manuscript.

*Competing interests.* The authors declare that they have no conflict of interest.

*Acknowledgments.* This work was financially supported by the National Natural Science Foundation of
China (91644213, 21925601) and the National Key R&D Program of China (2017YFC0209505). Lin
Wang acknowledges the Newton Advanced Fellowship (NA140106).



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





Table 1 Summary of experimental conditions.

| # | Precursor | Experimental condition | Precursor concentration (ppb) | Consumption of precursor (%) | RH (%) | Total flow rate (slpm) | $O_3$ concentration (ppb) |
|---|---|---|---|---|---|---|---|
| 1 | 1,2,4-TMB | OH | 158 | 59.3 | 12.5 | 10 | 712 |
| 2 | 1,3,5-TMB | OH | 118 | 62.8 | 13.6 | 10 | 845 |
| 3 | 1,2,3-TMB | OH | 214 | 58.4 | 8.1 | 10 | 1426 |
| 4 | 1,2,4-(1-methyl-D3)-TMB | OH | 155 | 62.0 | 11.6 | 10 | 1003 |
| 5 | 1,2,4-(2-methyl-D3)-TMB | OH | 169 | 61.8 | 12.5 | 10 | 776 |
| 6 | 1,2,4-(4-methyl-D3)-TMB | OH | 166 | 62.8 | 11.5 | 10 | 886 |
| 7 | 1,2,4-TMB | Low $NO_x$ (0.8ppb $NO_x$) | 170 | 61.5 | 12.7 | 10.4 | 944 |
| 8 | 1,2,4-TMB | Higher $NO_x$ (6.5ppb $NO_x$) | 145 | 69.7 | 9.3 | 10.4 | 3911 |





**Table 2.** Oxidation products of 1,2,4-TMB in categories of carbonyl, hydroxyl, and hydroperoxyl according to their molecular mass, as well as the potential peroxy radicals. Numbers in the parenthesis denote the relative intensity detected by Nitrate CI-APi-TOF in the OH-initiated oxidation of 1,2,4-TMB when that of the largest HOM signal ($C_9H_{16}O_8$) is arbitrarily set to be 100%. The relative intensity has been corrected with the relative transmission efficiency of Nitrate CI-APi-TOF.

| The potential peroxy radical $m$ | Carbonyl $m$-17 | Hydroxyl $m$-15 | Hydroperoxyl $m$+1 |
|---|---|---|---|
| $C_9H_{13}O_7\cdot$ | $C_9H_{12}O_6$ [a,b,c,d] (9.2 %) | $C_9H_{14}O_6$ [a,b,c,d] (20.3 %) | $C_9H_{14}O_7$ [b,c,d] (50.4 %) |
| $C_9H_{13}O_8\cdot$ | $C_9H_{12}O_7$ [b,c,d] (54.4 %) | $C_9H_{14}O_7$ [b,c,d] (50.4 %) | $C_9H_{14}O_8$ [c,d] (51.6 %) |
| $C_9H_{13}O_9\cdot$ | $C_9H_{12}O_8$ [d] (17.3 %) | $C_9H_{14}O_8$ [c,d] (51.6 %) | $C_9H_{14}O_9$ [d] (29.1 %) |
| $C_9H_{13}O_{10}\cdot$ | $C_9H_{12}O_9$ [d] (14.9 %) | $C_9H_{14}O_9$ [d] (29.1 %) | $C_9H_{14}O_{10}$ [d] (19.8 %) |
| $C_9H_{15}O_7\cdot$ | $C_9H_{14}O_6$ [a,b,c,d] (20.3 %) | $C_9H_{16}O_6$ [b,c,d] (2.3 %) | $C_9H_{16}O_7$ [b,c,d] (23.5 %) |
| $C_9H_{15}O_8\cdot$ | $C_9H_{14}O_7$ [b,c,d] (50.4 %) | $C_9H_{16}O_7$ [b,c,d] (23.5 %) | $C_9H_{16}O_8$ [c,d] (100 %) |
| $C_9H_{15}O_9\cdot$ | $C_9H_{14}O_8$ [c,d] (51.6 %) | $C_9H_{16}O_8$ [c,d] (100 %) | $C_9H_{16}O_9$ [d] (40.5 %) |
| $C_9H_{15}O_{10}\cdot$ | $C_9H_{14}O_9$ [d] (29.1 %) | $C_9H_{16}O_9$ [d] (40.5 %) | $C_9H_{16}O_{10}$ [d] (7.1 %) |

[a] These compounds are listed in the MCM mechanism of 1,2,4-TMB where they are formed by multiple OH oxidation steps.

[b] These compounds were detected by Vocus PTR.

[c] These compounds were detected by Iodide CI-TOF in both gas and particle phase.

[d] These compounds were detected by Nitrate CI-APi-TOF.



Table 3. Partially deuterated C9 products observed by Vocus PTR and/or Nitrate CI-APi-TOF. "V" and "N" denote observation by Vocus PTR and Nitrate CI-APi-TOF, respectively, whereas "-" means that the product was not observed by any instrument.

|  | 1,2,4-(1-methyl-D3)-TMB | 1,2,4-(2-methyl-D3)-TMB | 1,2,4-(4-methyl-D3)-TMB |
|---|---|---|---|
| $C_9H_{10}D_2O_6$ | V, N | - | - |
| $C_9H_{10}D_2O_7$ | V | - | - |
| $C_9H_{12}D_2O_8$ | - | - | N |





**Scheme Captions**

**Scheme 1.** A proposed autoxidation reaction scheme involving a bicyclic peroxy radical of $C_9H_{13}O_5\cdot$.

**Scheme 2.** A proposed autoxidation reaction scheme involving a bicyclic peroxy radical of $C_9H_{13}O_5\cdot$. Note that the reaction has been terminated with the formation of $C_9H_{14}O_4$ and re-initiated by a second OH attack.





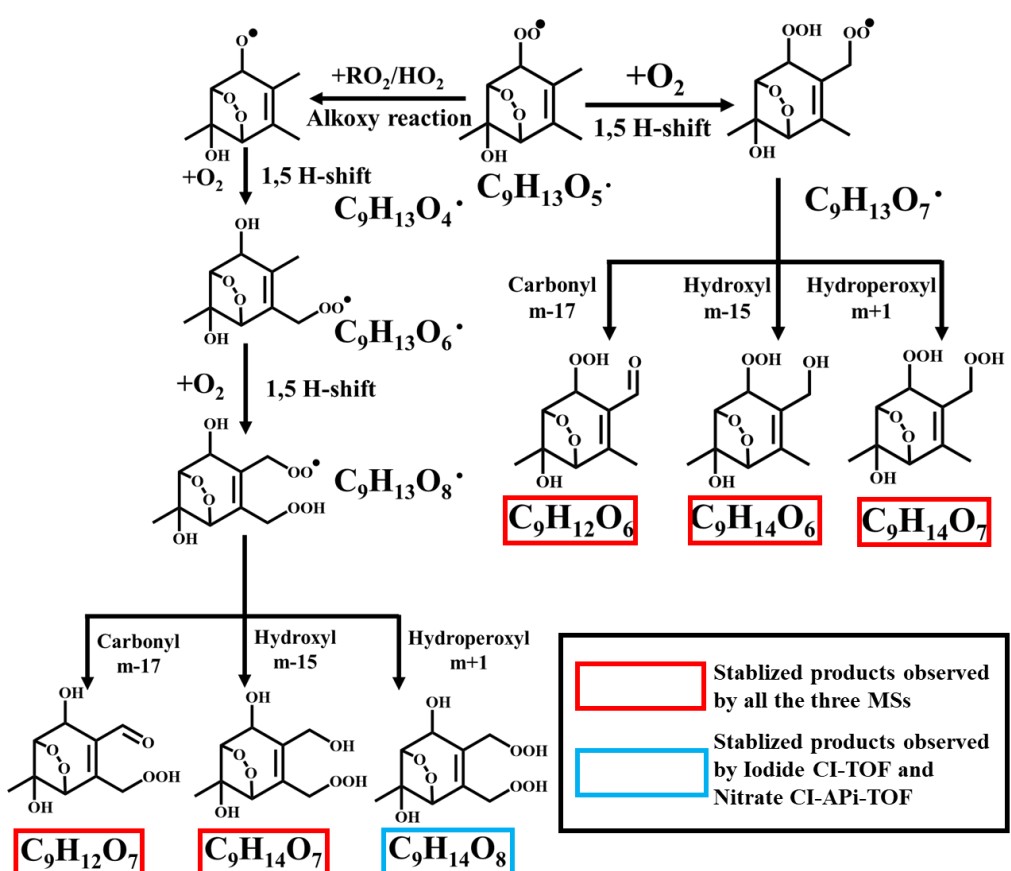

**Scheme 1**





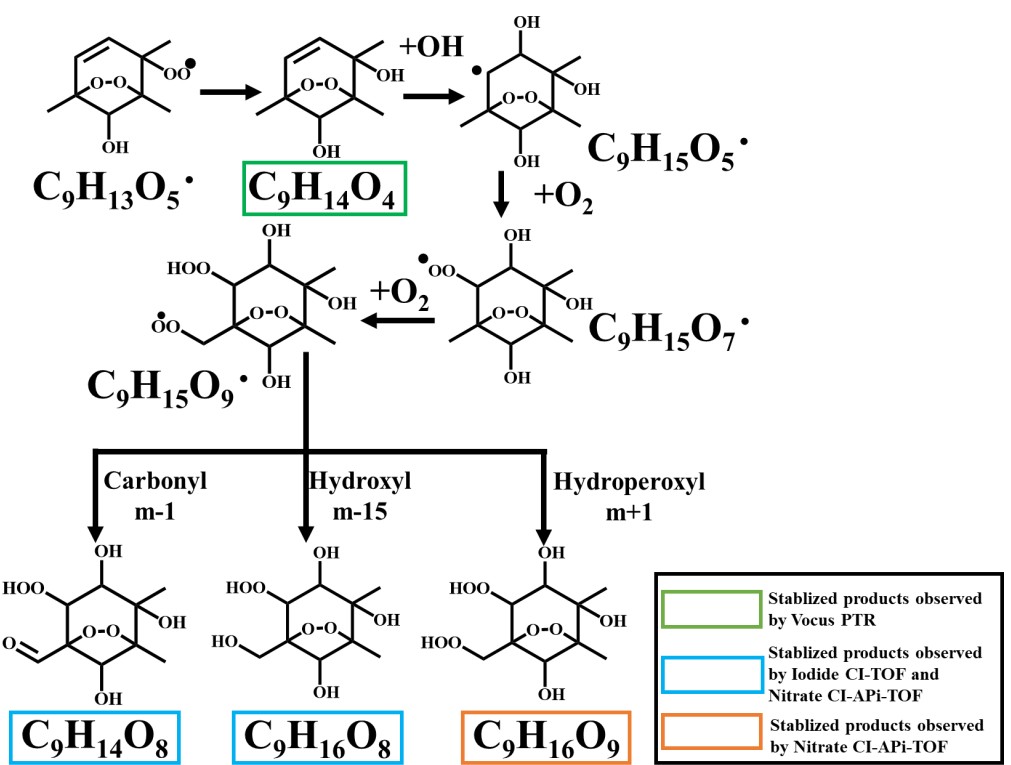

**Scheme 2**





**Figure Captions**

**Figure 1.** Schematics of experimental setup.

**Figure 2.** Comparison of C7-C9 products observed in the OH-initiated oxidation of 1,2,4-TMB (Exp. #1 in Table 1) with those listed in the MCM mechanism (Bloss et al., 2005). Filled red, orange, and green circles denote observation by Nitrate CI-APi-TOF, Iodide CI-TOF, and Vocus PTR, respectively, whereas open blue circles represent MCM species. The radius of filled circles are proportional to the signals of the compounds in each instrument. The signal of the most abundant product for each instrument is arbitrarily set to be 100%, but note that the arbitrary signals are not comparable among instruments. Symbols have been offset horizontally to avoid overlap.

**Figure 3.** Distribution of C9 products formed from OH-initiated reactions of TMBs (Exp. #1- 3 in Table 1) by (a) Vocus PTR, (b) Iodide CI-TOF for the gas phase, (c) Iodide CI-TOF for particle phase, and (d) Nitrate CI-APi-TOF. The yield of the most abundant product for each instrument is arbitrarily set to be 100%, but note that the arbitrary yields are not comparable among instruments. Also note that signal of Vocus PTR was processed in a logarithmic way before calculating the arbitrary yield.

**Figure 4.** (a) Distribution of $C_{18}H_{24}O_{8-13}$ and $C_{18}H_{26}O_{8-15}$ products formed from TMB oxidation experiments (Exp. #1-3 in Table 1), as measured by Nitrate CI-APi-TOF; (b) Distribution of $C_{18}H_{28}O_{9-15}$ and $C_{18}H_{30}O_{12-15}$ formed from TMB oxidation experiments (Exp. #1-3 in Table 1), as measure by Nitrate CI-APi-TOF; and (c) The total signal of C18 products formed from TMB oxidation experiments (Exp. #1-3 in Table 1), as measure by Nitrate CI-APi-TOF.

**Figure 5.** Relative contribution of C9 and C18 products formed from TMB oxidation experiments, as measured by Nitrate CI-APi-TOF. The relative intensity has been corrected with the relative transmission efficiency.

**Figure 6.** Comparison of C9 products detected by Nitrate CI-APi-TOF with zero, one or two nitrogen atoms formed from 1,2,4-TMB oxidation with different $NO_x$ settings.

**Figure 7.** (a) Comparison of C18 HOMs formed from 1,2,4-TMB oxidation with different $NO_x$ settings; and (b) Distribution of C18 organonitrates fomed from 1,2,4-TMB oxidation.



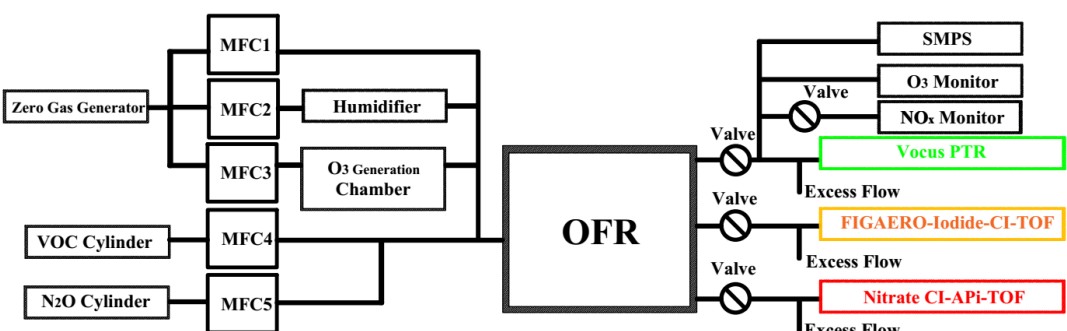

**Figure 1**





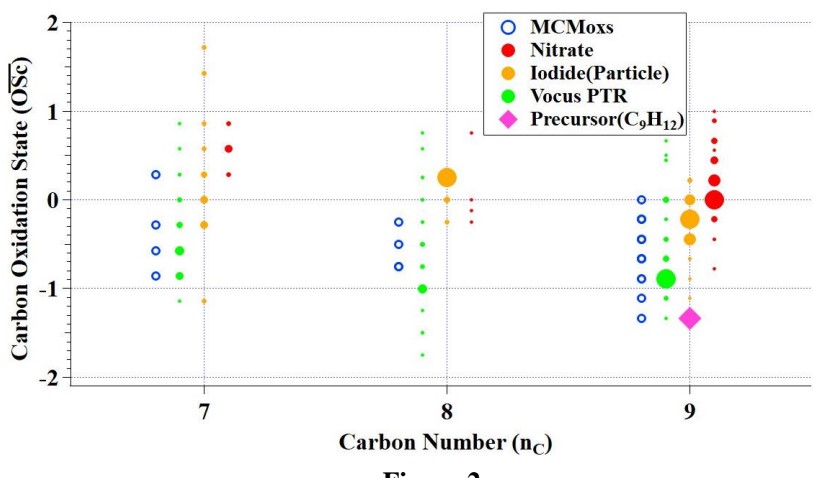

**Figure 2**



**Figure 3a**

**Figure 3b**

**Figure 3c**

**Figure 3d**

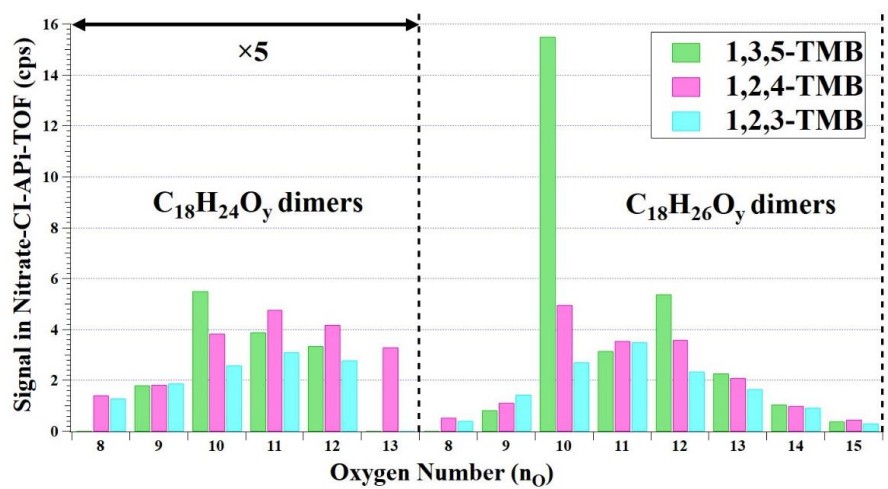

**Figure 4a**

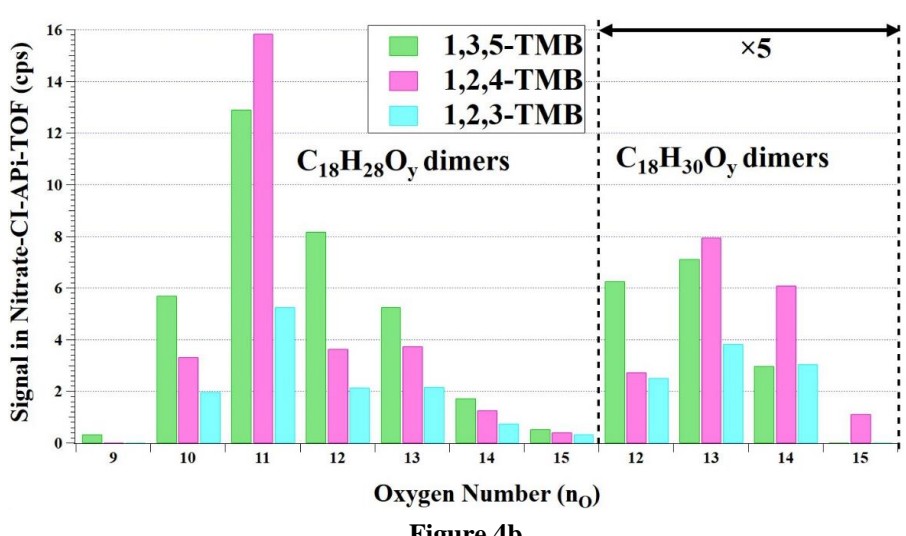

**Figure 4b**





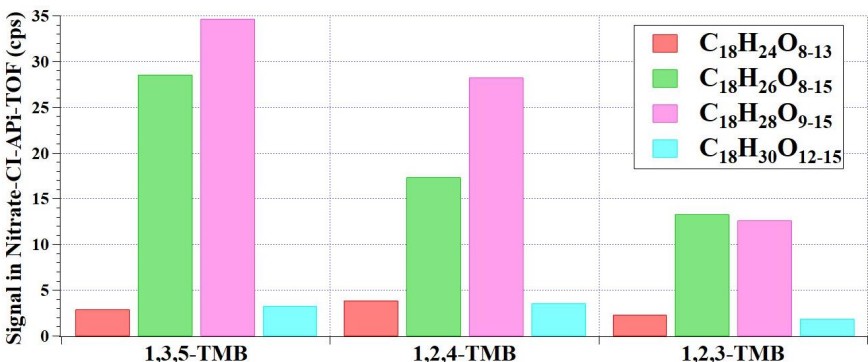

**Figure 4c**





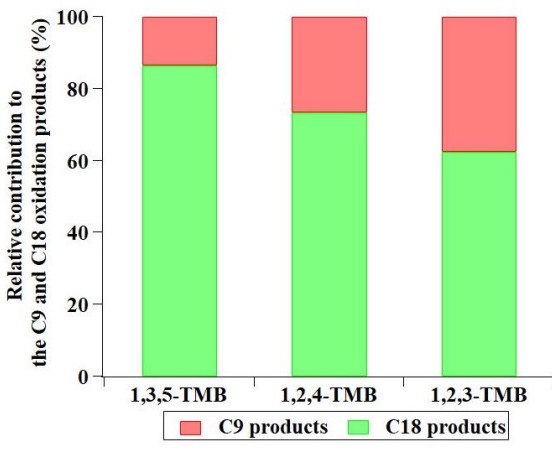

**Figure 5**





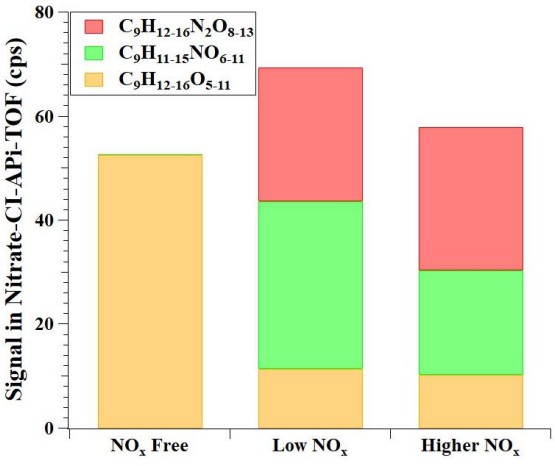

**Figure 6**





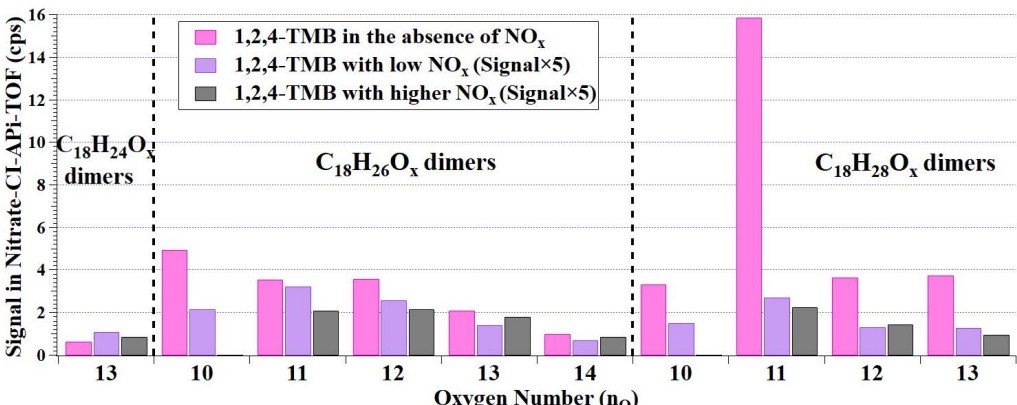

**Figure 7a**

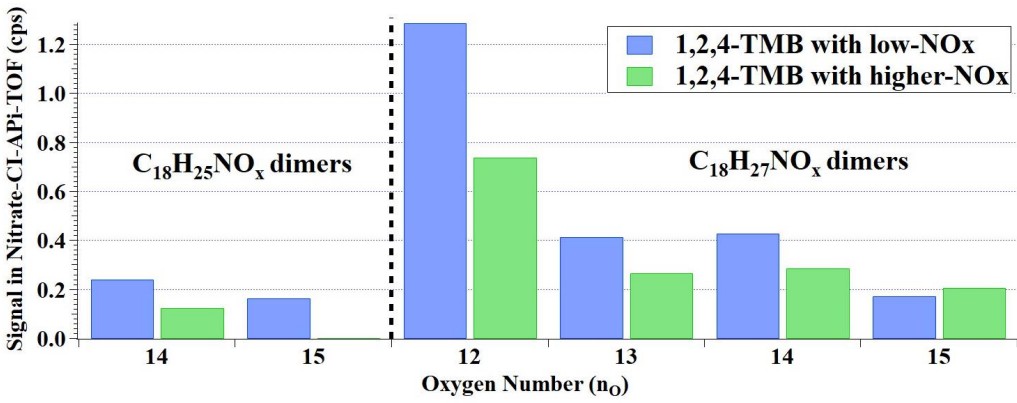

**Figure 7b**