# Peer review of "Oxygenated products formed from OH-initiated reactions of"

_Atmospheric Chemistry and Physics, 2020_

## Referee Comment (RC1) · Anonymous Referee #1 · 22 May 2020

GENERAL COMMENTS

In this manuscript the authors present results of an experimental study of the products of the reaction of trimethylbenzene isomers with OH radicals in the absence and presence of NOx. Reactions are conducted in a flow tube reactor and gas-phase products are analyzed using three types of online mass spectrometry that in combination allow detection of products ranging from very low to very high oxidation state. The authors detect a variety of monomer and dimer products that are not currently incorporated into the Master Chemical Mechanism, which is widely used to model the atmospheric chemistry of organic compounds. The authors propose structures for the compounds

(they are preliminary since only elemental formulas and deuterium labeling data are available) and mechanisms by which they could be formed. Fortunately, the authors have kept the paper concise, and not attempted to overinterpret the results or over-whelm readers with data. Although it is difficult to judge the importance of the results because of the lack of quantitative analysis, aromatics are an important class of compounds whose chemistry is not well understood, the work in technically sound, and it should be useful for others to build on. I think the paper should be published in ACP after the following minor comments and have been addressed.

SPECIFIC COMMENTS

1. Line 227-228: How can an oxidation product have an oxidation state lower than TMB?

2. Lines 260-262 (and elsewhere): Because of the high sensitivity of the nitrate-CIMS to oxidized products I do not think you can assume that products formed in these experiments did not involve multiple OH reactions. For example, in the work by Krechmer et al., EST (2016) they formed multigeneration products in chamber experiments in which the lights were only on for 10 s.

3. Line 325-326: The proposed isomerization of the bicyclic alkoxy radical would never compete with ring-opening pathways. See Vereecken and Peeters, PCCP (2009, 2010) estimation methods for these pathways.

4. Line 362: Please provide support (such as references) for the assumption of identical charging efficiencies. Given the high sensitivity of the nitrate-CIMS to oxidation state and chemical structure this sounds like a very poor assumption. I suspect this makes the fractions quoted in sections 3.3 and 3.4 misleading and possibly results is wrong conclusions. The only way I think one can deal with this is to always refer to these as a comparison of fractional signals or some such thing.

5. Line 407-409: Can you distinguish peroxynitrates by time profiles, since they should

decay by reversible decomposition on short timescales (though they may be reformed)?

TECHNICAL COMMENTS

1. Line 289: Should be "summarizes".

2. Line 345: Should be "more O atoms".

3. Line 362: Should replace "charge" with "charging" or "ionization".

4. Line 457: Delete "a".

5. Line 459: Should be "linearly dependent".
* * *

---

## Referee Comment (RC2) · Anonymous Referee #2 · 26 May 2020

MS No.: acp-2020-165

The authors describing experimental results of the OH radical initiated oxidation of different trimethylbenzenes carried out in a flow-through apparatus at atmospheric pressure in air. OH has been produced via 254 nm photolysis of ozone in presence of water vapour. Qualitative results of end product analysis are provided from three mass spec techniques. The authors chose relatively high initial reactant concentrations, $(3.5 - 5.2) \times 10^{12}$ molecules $cm^{-3}$, with a reactant conversion of 62.3 % each within the overall residence time of 80 s. Nothing is said regarding the $RO_2$ radical profiles in the experiments. It can be assumed from the stated reaction parameters that $RO_2$ levels are substantially higher than atmospheric. Consequently, especially $RO_2$ self- and cross reactions are favoured, which are of less importance for the $RO_2$ fate under atmospheric conditions. Thus, it´s not so surprising that a very big fraction of $C_{18}$ products has been detected. And that´s my main concern: Are the observed product distributions from these experiments relevant for the atmosphere? Some other points that should be considered:

- Line 137 – 141: Why did the authors take nitrous oxide as precursor for NO and $NO_2$? And again here: What is the NO and $NO_2$ profile in the experiments? One example from modelling should be given in the manuscript in order to allow the readership to get an impression for this.
- Line 197 – 203: From kinetic perspective, secondary chemistry, i.e. OH + product steps, cannot be neglected for a reactant conversion of 62.3 % in this system. What does it mean "one oxidation lifetime"?
- Table 1: The authors also used huge ozone concentrations in their runs, $(1.7 – 9.6) \times 10^{13}$ molecules $cm^{-3}$. After first OH attack the trimethylbenzene loses its aromaticity forming a series of unsaturated closed-shell products. What about the possible ozonolysis of these products?
- Schemes: It should be clarified what the authors mean with "stabilized products".
- Figure 5: A $C_{18}$ product fraction of more than 50% is very surprising for me. This finding should be discussed in respect of rates of the competing steps R5 – R8.

---

## Author Comment (AC1) · 23 Jun 2020

**RE: A point-to-point response to reviewers' comments**

"Oxygenated products formed from OH-initiated reactions of trimethylbenzene: Autoxidation and accretion" (acp-2020-165) by Yuwei Wang, Archit Mehra, Jordan E. Krechmer, Gan Yang, Xiaoyu Hu, Yiqun Lu, Andrew Lambe, Manjula Canagaratna, Jianmin Chen, Douglas Worsnop, Hugh Coe, Lin Wang

Dear Prof. Dr. Markus Ammann,

We are very grateful to the helpful comments from the reviewers, and have carefully revised our manuscript accordingly. A point-to-point response to the comments, which are repeated in italic, is given below.

We are looking forward to the reviewers' feedback and your decision with the revision.

Best regards,

Lin Wang
Fudan University
lin_wang@fudan.edu.cn

**Reviewer #1**

**GENERAL COMMENTS**

*In this manuscript the authors present results of an experimental study of the products of the reaction of trimethylbenzene isomers with OH radicals in the absence and presence of NOx. Reactions are conducted in a flow tube reactor and gas-phase products are analyzed using three types of online mass spectrometry that in combination allow detection of products ranging from very low to very high oxidation state. The authors detect a variety of monomer and dimer products that are not currently incorporated into the Master Chemical Mechanism, which is widely used to model the atmospheric chemistry of organic compounds. The authors propose structures for the compounds (they are preliminary since only elemental formulas and deuterium labeling data are available) and mechanisms by which they could be formed. Fortunately, the authors have kept the paper concise, and not attempted to overinterpret the results or overwhelm readers with data. Although it is difficult to judge the importance of the results because of the lack of quantitative analysis, aromatics are an important class of compounds whose chemistry is not well understood, the work in technically sound, and it should be useful for others to build on. I think the paper should be published in ACP after the following minor comments and have been addressed.*

**Response**. We are very grateful to the positive viewing of our manuscript by Reviewer #1, and have now revised our manuscript accordingly.

**SPECIFIC COMMENTS**

1. *Line 227-228: How can an oxidation product have an oxidation state lower than TMB?*

**Response**. In the original manuscript, these products refer to ions of $C_8H_{15}O^+$, $C_8H_{15}O_2^+$, $C_8H_{17}O^+$, and $C_8H_{17}O_2^+$ detected only by Vocus PTR, all of which have very low signal intensities ranging from 14.67 cps to 78.47 cps. As a comparison, the ion counts for the most abundant C9 products ($C_9H_{11}O^+$) and C8 products ($C_8H_{11}O^+$) detected by Vocus PTR were 13887 cps and 5023 cps, respectively. The concentrations of these ions should be quite low, considering the high sensitivity of Vocus PTR to compounds with a few oxygen atoms (around 8000 cps/ppb to xylene and TMB, and 8500 cps/ppb to methyl ethyl ketone as calibrated with a commercial calibration cylinder). At the same time, there is a documented history of fragmentation during PTR ionization, leading to a leakage of oxygen atom(s) (de Gouw and Warneke, 2007; Gueneron et al., 2015; Karl et al., 2018; Tani, 2013; Yuan et al., 2017). Though oxygenated VOCs are reported to be less significantly fragmented than alkanes and many alkenes (Yuan et al., 2017), a leakage of oxygen due to fragmentation is typically on the order of < 5% for ketones, 15% for aldehydes, < 5% for ethers, <10% for carboxylic acids, 30% for peroxides, 40% for diols, and 70% for alcohols (Karl et al., 2018; Španěl and Smith, 2013). Hence, we postulate that these ions might come from fragmentation of parent compounds in the FIMR (focusing ion-molecule reactor) of Vocus PTR. We have removed this statement in Line 225-228 of the original manuscript and updated Figure 2.

The revised Figure 2 is shown below.

[Figure]

**Figure 2**

2. *Lines 260-262 (and elsewhere): Because of the high sensitivity of the nitrate-CIMS to oxidized products I do not think you can assume that products formed in these experiments did not involve multiple OH reactions. For example, in the work by Krechmer et al., EST (2016) they formed multigeneration products in chamber experiments in which the lights were only on for 10 s.*

**Response**. We agree with this reviewer that multiple OH attacks can occur in our reaction system, which is evidenced by the observation of $C_9H_{16}O_{6-9}$ products, as 16 hydrogen atoms in the molecular formula can be regarded as a characteristic of the second-generation products according to Molteni et al. (2018). In fact, we mentioned the possibility of multiple OH-attacks in multiple places in our manuscript, i.e., statements in Line 238 of the original manuscript and Table 2.

In Line 260-262 of the original manuscript, we intended to state that $C_9H_{12}O_6$ as a multi-generation hydroperoxyl product as predicted by MCM should not have a comparable yield as that formed through the carbonyl termination reaction of a peroxy radical that is involved in autoxidation, since a multi-generation product is not favored at OH exposure as short as one life time of TMB. $C_9H_{14}O_6$ is another product predicted by MCM but we missed it in main text of the original manuscript. $C_9H_{14}O_6$ is unlikely to be formed with a considerable yield through the MCM route for a similar reason.

To avoid misunderstanding, we have revised our manuscript, which (Line 268-275) reads "$C_9H_{12}O_6$ is one of the only two signals that have been predicted by MCM,…, which is unlikely to contribute a lot to the observed signal of $C_9H_{12}O_6$ since the concentration of a multi-generation product is not expected to be high at OH exposure as short as one lifetime of TMB. $C_9H_{14}O_6$ is the other one, presumed to be a hydroperoxyl product of a second-generation peroxy radical formed via the epoxy-oxy pathway (MCM name: TM124MUOOH), which is unlikely to be formed through the MCM route with a considerable yield, either."

Please also refer to our response to Comment #3 from Reviewer #2.

3. *Line 325-326: The proposed isomerization of the bicyclic alkoxy radical would never compete with ring-opening pathways. See Vereecken and Peeters, PCCP (2009, 2010) estimation methods for these pathways.*

**Response**. Thanks for pointing out this issue. After a detailed discussion with Dr. Vereecken, quote, "For this structure, consider that the H-migration is across a trans-substituted alkene: the $CH_3$-C=C-CHO· carbons are all in one plane. Furthermore, because of the ring structure, the CHO· group cannot rotate the oxygen towards the $CH_3$ H-atom. Hence, to shift, the H-atom would have to "leap" almost 5 angstroms without being attached to anything, whereas normally it is less than 1.4 angstroms from either or both of the starting-C/ending-O atom", we agree with reviewer and Dr. Vereecken that the 1, 5-H-shift of this alkoxy radical is virtually impossible, and have deleted this pathway in the revised manuscript.

We have updated Scheme 1 as shown below, where the formation of $C_9H_{12}O_6$, $C_9H_{14}O_6$, and $C_9H_{14}O_7$ was proposed with the involvement of an autoxidation step.

[Figure]

**Scheme 1**

The text in Line 324-327 of the original manuscript have been deleted. We now state in the revised manuscript that (Line355-358)"An autoxidation reaction pathway that can explain the observation of $C_9H_{10}D_2O_7$ in the 1,2,4-(1-methyl-D3)-TMB + OH experiment is currently unavailable, although we speculate that a "peroxy-alkoxy-peroxy" conversion is likely involved during the formation of $C_9H_{12}O_7$ according to the number of oxygen atoms" .

4. *Line 362: Please provide support (such as references) for the assumption of identical charging efficiencies. Given the high sensitivity of the nitrate-CIMS to oxidation state and chemical structure this sounds like a very poor assumption. I suspect this makes the fractions quoted in sections 3.3 and 3.4 misleading and possibly results is wrong conclusions. The only way I think one can deal with this is to always refer to these as a comparison of fractional signals or some such thing.*

**Response**. We believe that this assumption is the best option when there are no real measurements. Hyttinen et al. (2015) modelled the charging of highly oxygenated products from cyclohexene ozonolysis using nitrate-CIMS, showing a similar charging efficiency of nitrate source for highly oxygenated compounds. Ehn et al. (2014) assumed that the nitrate source has the same sensitivity for all highly oxygenated molecules. We agree with this reviewer that nitrate-CIMS is quite sensitive to the oxidation state of compounds to be measured. However, once a compound is highly oxygenated (i.e., contains 6 or more oxygen atoms) or has at least two hydrogen bond donor functional groups (for example, hydroperoxide, OOH), it can be assumed to be charged at the collision limit (Ehn et al, 2014; Hyttinen et al., 2015). Clearly, HOM monomer and dimer contain more than two hydrogen bond donor functional groups or 6 or more oxygen atoms. Though dimers possess more functional groups that favor the binding with $NO_3^-$ than monomers do, the charging efficiency cannot be higher than collision limit.

We have cited the above references to support our assumption, which (Line 379-380) reads "The charging efficiency for C9 and C18 products is assumed to be identical in Nitrate CI-APi-TOF (Ehn et al., 2014; Hyttinen et al., 2015)".

5. *Line 407-409: Can you distinguish peroxynitrates by time profiles, since they should decay by reversible decomposition on short timescales (though they may be reformed)?*

**Response**. This is a good point. However, during our experimental procedure, we focused more on the establishment of stable signals of key products, which was determined by multiple factors including wall loss, and did not try to tackle the stability of products by terminating the reaction and monitoring the decay of signals. Thus, a comparison of time profiles of various nitrogen-containing products is currently not available.

TECHNICAL COMMENTS
1. *Line 289: Should be "summarizes".*
**Response**. We have replaced "summaries" with "summarizes".

2. *Line 345: Should be "more O atoms".*
**Response**. We have revised our manuscript accordingly.

3. Line 362: Should replace "charge" with "charging" or "ionization".
**Response**. We have replaced "charge" with "charging".

4. *Line 457: Delete "a".*
**Response**. We have removed this "a".

5. *Line 459: Should be "linearly dependent".*
**Response**. We have revised the text as "linearly dependent on".

**Reviewer #2**

*The authors describing experimental results of the OH radical initiated oxidation of different trimethylbenzenes carried out in a flow-through apparatus at atmospheric pressure in air. OH has been produced via 254 nm photolysis of ozone in presence of water vapour. Qualitative results of end product analysis are provided from three mass spec techniques. The authors chose relatively high initial reactant concentrations, $(3.5 – 5.2) \times 10^{12}$ molecules $cm^{-3}$, with a reactant conversion of 62.3 % each within the overall residence time of 80 s. Nothing is said regarding the $RO_2$ radical profiles in the experiments. It can be assumed from the stated reaction parameters that $RO_2$ levels are substantially higher than atmospheric. Consequently, especially $RO_2$ self- and cross reactions are favoured, which are of less importance for the $RO_2$ fate under atmospheric conditions. Thus, it´s not so surprising that a very big fraction of C18 products has been detected. And that´s my main concern: Are the observed product distributions from these experiments relevant for the atmosphere?*

**Response**. Indeed, the initial reactant concentrations are much higher than those under atmospheric conditions. However, high concentrations of VOCs were deliberately chosen in this study. We aim to experimentally observe highly oxygenated products to confirm the possibility of autoxidation, and to propose the detailed autoxidation pathways via the comparison between reactions of un-deuterated and partially deuterated reactants. High concentrations of reactants will certainly help identify the highly oxygenated products that are of low volatility and easy to loss. At the same time, we did not over-interpret our results by hinting that the observed product distributions from the experiments are the same as those in the ambient atmosphere. Our viewpoints in section 3.3, where the characteristics of C18 products are discussed, are (1) to confirm the extensive existence of highly oxygenated $RO_2$ radicals, in other words, the extensive existence of the autoxidation pathways in the OH-initiated oxidation of TMB as an echo of the last sentence in section 3.1; (2) to provide an evidence on the structural enhancement in accretion product formation. In summary, what we have focused on is the formation mechanism and chemical fates of the $RO_2$ radicals and HOM products.

To clarify this point, we now state in our revised manuscript (Line 217-220) that "Also note that the concentrations of precursors in our experiments were much higher than the atmospheric ones. These concentrations were deliberately chosen to help identify the highly oxygenated products that are of low volatility and easy to loss in the sampling, but subject to the side effect that the relative significance of different pathways could be altered"

and Line (423-425) that "Again, it should be noted that this result was obtained under the condition of very high concentrations of precursors and thus the relative fractions of products could be different under ambient conditions".

*Some other points that should be considered:*
1. *- Line 137 – 141: Why did the authors take nitrous oxide as precursor for NO and $NO_2$?*

**Response**. Parts-per-million (ppm) levels of $O_3$ are required to generate OH radicals, which prevent sustained $NO_x$ (especially NO) mixing ratios at sufficient levels to compete with $HO_2$ as a sink for $RO_2$, due to the fast conversion of $NO_x$ to nitric acid ($HNO_3$) via the reactions of $NO+O_3 \rightarrow NO_2+O_2$ and $NO_2+OH \rightarrow HNO_3$. On the other hand, $N_2O$ is a better $NO_x$ precursor specifically in OFR studies for the following reasons, as described in Lambe et al. (2017) and Peng et al. (2018), and recently reviewed in Peng et al. (2020):

1) The spatial distribution of NO and $NO_2$ generated via the $N_2O + O(^1D)$ reaction is more homogenous than what is achieved by simple additions of NO and/or $NO_2$, because of the continuous production of $O(^1D)$ from the $O_3$ photolysis inside the reactor.

2) Steady-state mixing ratios of NO from $O(^1D) + N_2O$ reactions are orders of magnitude higher than that from a simple NO injection

*2. And again here: What is the NO and NO2 profile in the experiments? One example from modelling should be given in the manuscript in order to allow the readership to get an impression for this.*

**Response**. Thanks for this suggestion. We utilized a photochemical model (PAM_chem_v8) (Lambe et al., 2017; Li et al., 2015; Peng et al., 2015) to investigate the NOx concentrations and $NO/NO_2$ profiles in the OFR. Unfortunately, the $NO/NO_2$ concentrations in the model output (tens of ppb) are much larger than our reported values (a few ppb) in our Table 2 of the original manuscript, whereas the comparison of $[O_3]$ and [1,2,4-TMB] between modelled values and measured ones looks fine (Figures R1 and R2).

[Figure]

**Figure R1.** Modelled profiles by PAM_chem_v8 of different oxidants, $NO_x$ and the precursor under the settings of "low $NO_x$ experiment" (initial $[O_3]$ = 1.8 ppm, initial [1,2,4-TMB] = 170 ppb, and irradiance of

254 nm Lamps = $2.0 \times 10^{15}$ ph cm$^{-2}$ s). The measured [O$_3$] and [1,2,4-TMB] at the exit of OFR are shown by a triangle and a diamond, respectively, in the plot. The vertical purple line represents a residence time of 77.3 s."

[Figure]

**Figure R2.** Modelled profiles by PAM_chem_v8 of different oxidants, NO$_x$ and the precursor under the settings of "higher NO$_x$ experiment" (initial [O$_3$] = 6.7 ppm, initial [1,2,4-TMB] = 145 ppb, and irradiance of 254 nm Lamps = $1.28 \times 10^{15}$ ph cm$^{-2}$ s). The measured [O$_3$] and [1,2,4-TMB] at the exit of OFR are shown by a triangle and a diamond, respectively, in the plot. The vertical purple line represents a residence time of 77.3 s."

To investigate this discrepancy, we recalibrated our NO$_x$ monitor and performed a series of new experiments under experimental conditions similar to experiments #7 and #8 (Table R1), and generally observed a factor of two discrepancy between modelled and measured NO/NO$_2$ (Figure R3 and R4). The mean ratio of modelled-to-measured [NO] at the exit of OFR were $0.64 \pm 0.04$ and $0.98 \pm 0.01$ for Exp. #R1-R4 (low NO$_x$) and Exp. #R5-R8 (higher NO$_x$), respectively, whereas those of modelled-to-measured [NO$_2$] at the exit of OFR were $0.51 \pm 0.07$ and $0.73 \pm 0.01$ for Exp. #R1-R4 and Exp. #R5-R8, respectively. Hence, we reached a conclusion that our NO$_x$ monitor malfunctioned during our previous experiments. Since the setting of mass spectrometers have altered significantly and thus the new mass spectrometric results are not directly comparable to those in previous experiment. We decide to keep the previous mass spectrometric results but report the modelled NO$_x$ concentrations, which have no impacts on the conclusions of this study.

**Table R1.** Summary of experimental conditions for a series of new experiments carried out during the revision. The total flow was set to be 10.4 slpm and [N$_2$O] was the same as that in the previous experiments.

Exp. #R1-R4 correspond to Exp. #7 (low $NO_x$), and Exp. #R5-R8 correspond to Exp. #8 (high $NO_x$). In Exp. #R1-R4, the 254 nm lamps were tuned to get different $NO/NO_2$ levels and so were in Exp. #R5-R8. Reported [NO] and [$NO_2$] are values at the exit of OFR.

| # | Precursor | Precursor concentration (ppb) | Consumption of precursor (%) | RH(%) | $O_3$ concentration (ppb) | Measured $NO(ppb)/NO_2(ppb)$ | Modelled $NO(ppb)/NO_2$ (ppb) |
|---|---|---|---|---|---|---|---|
| R1 | 1,2,4-TMB | 182 | 59.2 | 15.2 | 990 | 3.8/336.5 | 2.5/152.1 |
| R2 | 1,2,4-TMB | 191 | 51.7 | 15.2 | 1185 | 2.03/244.9 | 1.6/113.2 |
| R3 | 1,2,4-TMB | 205 | 60.2 | 15 | 868 | 5.38/360.6 | 3.2/185.0 |
| R4 | 1,2,4-TMB | 220 | 41.5 | 14.9 | 1342 | 1.17/178.1 | 1.1/84.1 |
| R5 | 1,2,4-TMB | 141 | 51.3 | 8.7 | 3953 | 3.2/791.8 | 3.1/550.9 |
| R6 | 1,2,4-TMB | 147 | 46.7 | 8.7 | 4671 | 1.8/528.2 | 1.7/365.1 |
| R7 | 1,2,4-TMB | 150 | 52.0 | 8.7 | 3375 | 4.5/1005.5 | 4.5/692.0 |
| R8 | 1,2,4-TMB | 155 | 52.9 | 8.5 | 2946 | 6.1/1079.9 | 6.0/854.1 |

[Figure]

**Figure R3.** Measured *v.s.* modelled [NO] at a residence time of 77.3 s at the exit of the OFR. Error bars represent either ± 60% uncertainty in model outputs (Peng et al., 2015) or ± 10% precision in [NO] measurements by a calibrated NO$_x$ monitor. The mean ratio of modelled-to-measured [NO] at the exit of OFR were 0.64 ± 0.04 and 0.98 ± 0.01 for Exp. #R1-R4 and Exp. #R5-R8, respectively.

[Figure]

**Figure R4.** Measured *v.s.* modelled [NO$_2$] at a residence time of 77.3 s at the exit of the OFR. Error bars represent either ± 60% uncertainty in model outputs (Peng et al., 2015) or ± 20% precision in [NO$_2$] measurements by a calibrated NO$_x$ monitor. The mean ratio of modelled-to-measured [NO$_2$] at the exit of OFR were 0.51 ± 0.07 and 0.73 ± 0.01 for Exp. #R1-R4 and Exp. #R5-R8, respectively.

We have now stated in our revised manuscript (Line 119-121) that "In addition, an ozone monitor (Model 106-M, 2B technologies) was utilized to measure ozone concentration, whereas a set of …",

(Line 127) that "approximately 80 seconds (77.3 seconds at 10.4 slpm)",

(Line 143-145) that "A photochemical model (PAM_chem_v8) (Lambe et al., 2017; Li et al., 2015; Peng et al., 2015) was implemented to constrain the NO/NO$_2$ profiles in the experiments, whose details are presented in Section S1",

(Line 428-432) that "To constrain the NO$_x$ level in the OFR, the profiles of NO/NO$_2$ were modelled by PAM_chem_v8, as shown in Figure S5. The mathematically-averaged NO$_x$ levels in the low NO$_x$

experiment (Exp. #7) and higher $NO_x$ experiment (Exp. #8) were 92 ppb (2.5 ppb NO + 89.5 ppb $NO_2$) and 295.3 ppb (2.9 ppb NO + 292.4 ppb $NO_2$), respectively. The $NO_x$/VOC in our experiments is comparable to ambient values in polluted areas. The $NOx/(\Delta VOC)$ was around 0.9 in the low $NO_x$ experiment and 2.9 in the higher $NO_x$ one."

and (Section S1 in the supplement) that "Figure S5 shows the modelled profiles of the major oxidants, $NO_x$, and the precursor under the settings of Exp. #7 and Exp. #8 in Table 1. In the low $NO_x$ experiment, the modelled $[O_3]$ is 20% lower than the measured value at the exit of OFR whereas the modelled [1,2,4-TMB] is 19% higher than the measured one. In the higher $NO_x$ experiment, the modelled $[O_3]$ is 3% higher than the measured value whereas the modelled [1,2,4-TMB] is 23% higher than the measured one.

(a)

[Figure]

(b)

[Figure]

**Figure S5.** Modelled profiles by PAM_chem_v8 of different oxidants, $NO_x$ and the precursor under the settings of (a) low $NO_x$ experiment (initial $[O_3]$ = 1.8 ppm, initial $[1,2,4\text{-TMB}]$ = 170 ppb, and irradiance of 254 nm Lamps = $2.0 \times 10^{15}$ ph cm$^{-2}$ s), and (b) higher $NO_x$ experiment (initial $[O_3]$ = 6.7 ppm, initial $[1,2,4\text{-TMB}]$ = 145 ppb, and irradiance of 254 nm Lamps = $1.28 \times 10^{15}$ ph cm$^{-2}$ s). The measured $[O_3]$ and $[1,2,4\text{-TMB}]$ at the exit of OFR are shown by a triangle and a diamond in the plot. The vertical purple line represents a residence time of 77.3 s.

The updated Table 1 is shown below

**Table 1.** Summary of experimental conditions.

| # | Precursor | Experimental condition | Precursor concentration (ppb) | Consumption of precursor (%) | RH (%) | Total flow rate (slpm) | $O_3$ concentration (ppb) |
|---|-----------|------------------------|-------------------------------|------------------------------|--------|------------------------|----------------------------|
| 1 | 1,2,4-TMB | OH | 158 | 59.3 | 12.5 | 10 | 712 |
| 2 | 1,3,5-TMB | OH | 118 | 62.8 | 13.6 | 10 | 845 |
| 3 | 1,2,3-TMB | OH | 214 | 58.4 | 8.1 | 10 | 1426 |
| 4 | 1,2,4-(1-methyl-D3)-TMB | OH | 155 | 62.0 | 11.6 | 10 | 1003 |
| 5 | 1,2,4-(2-methyl-D3)-TMB | OH | 169 | 61.8 | 12.5 | 10 | 776 |

| 6 | 1,2,4-(4-methyl-D3)-TMB | OH | 166 | 62.8 | 11.5 | 10 | 886 |
| 7 | 1,2,4-TMB | Low NO$_x$ (2.5 ppb NO + 89.5 ppb NO$_2$) [a] | 170 | 61.5 | 12.7 | 10.4 | 944 |
| 8 | 1,2,4-TMB | Higher NO$_x$ (2.9 ppb NO + 292.4 ppb NO$_2$) [a] | 145 | 69.7 | 9.3 | 10.4 | 3911 |

[a] Modelled mathematically-averaged NO/NO$_2$ concentrations in the OFR are shown here because of the malfunction of a NO$_x$ monitor. The model underestimates [NO] and [NO$_2$] by up to a factor of 2, according to separate experiments that are not presented.

Lastly, the text in the Line 403-405 of the original manuscript that "The NO$_x$ levels in the low NO$_x$ experiment (Exp. #7) and higher NO$_x$ experiment (Exp. #8) were 0.8 ppb and 6.5 ppb, respectively. Compared to the ambient values in polluted areas, this NO$_x$/VOC is low. The NO$_x$/($\Delta$VOC) was around 0.8% in the low NOx experiment and 6.4% in higher NO$_x$ one" have been deleted.

*3. - Line 197 – 203: From kinetic perspective, secondary chemistry, i.e. OH + product steps, cannot be neglected for a reactant conversion of 62.3 % in this system. What does it mean "one oxidation lifetime"?*

**Response**. The statement "under this condition, the production of the first-generation products is generally favored" does not exclude the possibility of secondary reactions. "One oxidation lifetime" means the concentration of the reactant decreases to 1/*e* of its initial concentration, i.e., consumption of 63.2% of the initial reactant. To clarify our points, we have now stated in our revised manuscript that (Line 204-208) "…so that the OH exposure in the OFR was close to one oxidation lifetime of TMB, i.e., consumption of (1-1/*e*) of the initial TMB. Under this condition, the production of the first-generation products is generally favored and the multi-generation products are also present, if the subsequent loss reactions for these products are assumed to proceed in the similar rate."

Please also refer to our response to Comment #2 from Reviewer #1.

*4. -Table 1:The authors also used huge ozone concentrations in their runs, (1.7 – 9.6) ×10$^{13}$ molecules cm$^{-3}$. After first OH attack the trimethylbenzene loses its aromaticity forming a series of unsaturated closed-shell products. What about the possible ozonolysis of these products?*

**Response**. The significance of ozonolysis of these products can be estimated using the concentrations of ozone and OH, and the associated reaction rates with unsaturated closed-shell products. The decay of a VOC by reactions with OH/O$_3$ is defined as,

$$\frac{d[VOC]}{dt} = -k_{VOC+oxidant}[\text{Oxidant}]\,[VOC]$$

where $k_{VOC+oxidant}$ is the reaction rates of VOC with OH/O₃. As a result, the lifetime of a VOC due to its reactions with OH/O₃ is defined as,

$$t_{OH} = \frac{1}{k_{VOC+oxidant}[\text{Oxidant}]}$$

Since the concentration of TMB can be measured online via Vocus PTR, we can determine the averaged OH radical concentration in the oxidation flow reactor according to the decay of TMB, whereas the concertation of $O_3$ can be measured with an $O_3$ box.

Now let's take the 1,3,5-TMB + OH experiment for example, where [OH] = $2.2×10^8$ molecules cm$^{-3}$, and [O₃] = $2.11×10^{13}$ molecules cm$^{-3}$. No systematic research has been performed on the reaction rate of ozone with the first-generation oxidation products of trimethylbenzene. However, the reaction rates for ozonolysis of VOCs typically range from $10^{-16}$ to $10^{-18}$ cm$^3$ molecules$^{-1}$ s$^{-1}$. A distinct feature of the aromatic oxidation is the faster oxidation rates of the first-generation products as compared to the parent molecule (Garmash et al., 2020). The pi-electron structure of the aromatic ring makes the parent molecule less susceptible towards OH oxidation compared to most organic molecules. It is thus assumed that the reaction rate between OH and the typical bicyclic first-generation product is ~$1.5×10^{-10}$ molecules cm$^{-3}$ (MCM v3.3.1, available at: http://mcm.leeds.ac.uk/MCM). Therefore, we can determine the lifetimes of unsaturated closed-shell products against OH and ozone, respectively.

$$t_{OH} = 30.3 \text{ s}$$
$$t_{O_3} = 473.9 \sim 47393.3 \text{ s}$$

Hence, ozonolysis is not expected to be a significant reaction route in our system, which is consistent with a former study (Molteni et al., 2018). Also, Berndt et al. (2018) show that the formation of ROOR' accretion products from TMB is a pure RO₂+R'O₂ gas-phase reaction without any hidden effects exerted by ozonolysis.

*5. - Schemes: It should be clarified what the authors mean with "stabilized products".*

**Response**. Here, the term "stabilized products" refers to "non-radical products", which shows up for a lot of times in the manuscript. On the other hand, several researchers used "closed-shell products", which we prefer not to use. We have revised our manuscript and added a clarification at the position where this term shows up for the first time (Line 96 - 97):" Subsequent reactions of the intermediates will lead to the formation of stabilized products (or non-radical products),"

*6. - Figure 5: A C₁₈ product fraction of more than 50% is very surprising for me. This finding should be discussed in respect of rates of the competing steps R5 – R8.*

**Response**. In fact, our study is not the first one that observed a dimer product fraction of more than 50%, as mentioned in Line 377-379 from the previous version of manuscript. It is likely a result from the high

precursor concentrations and should not be implied to the ambient as discussed in our response to the general comment from Reviewer #2.

We have now stated in our manuscript that (Line 393-417) "In the 1,3,5-TMB oxidation experiments (Exp. #2), where the highest C18 dimer fraction was observed, the mole fraction of the C18 dimers is likely determined by the competition of reactions R5, R6, R7, and R8, which can be mathematically expressed as

$$f_{C18} = \frac{0.5 \times k_{R8}[RO_2]}{k_{R5,R6}[RO_2] + k_{R7}[HO_2] + 0.5 \times k_{R8}[RO_2]} \qquad (1)$$

where $k_{R5,R6}$ stands for the reaction rates for R5 and R6, assumed to be around 8.8 $\times 10^{-13}$ cm$^3$ molecule$^{-1}$ s$^{-1}$ by MCM, $k_{R7}$ is the reaction rate for R7, set at a typical value of 2 $\times 10^{-11}$ cm$^3$ molecule$^{-1}$ s$^{-1}$ (Berndt et al., 2018b; Bianchi et al., 2019), and $k_{R8}$ is the reaction rate of R8 for BPRs generated by 1,3,5-TMB, which has recently been measured to be as fast as $10^{-10}$ cm$^3$ molecule$^{-1}$ s$^{-1}$ (Berndt et al., 2018b).

Since the concentration of HO$_2$ in the OFR was not measured, we utilized a kinetic reaction model (PAM_chem_v8) to characterize the concentration profiles of oxidants in the OFR, which include OH, O$_3$, HO$_2$, and H$_2$O$_2$. A detailed description of this model is given in section S1 of the supplement and the modelled profiles of oxidants and precursors are shown in Figure S4. According to the model, the steady-state concentration of HO$_2$ in the Exp. #2 was around 18 ppt (~ 4.5 $\times$ 10$^8$ molecules cm$^{-3}$). On the other hand, it is difficult to evaluate the effective concentration of the RO$_2$ radicals in the system, because RO$_2$ with low oxidation states will not form HOMs via reactions R5-R8. Therefore, we estimated the concentration of RO$_2$ in Eq. (1) to be close to that of BPRs in the OFR. According to MCM v3.3.1, the branching ratio for the peroxide-bicyclic pathway in the OH oxidation of 1,3,5-TMB is 79%, so that the concentration of BPRs was roughly estimated to be 58.5 ppb (~ 1.5 $\times$ 10$^{12}$ molecules cm$^{-3}$, 79% of the reacted 1,3,5-TMB). Hence, the fraction of C18 dimer is estimated to be around 98%. Clearly, this estimation itself comes with a large uncertainty, and the estimated fraction can only be regarded as an indication of explainable high yields of C18 dimers instead of a rigorous number.

In fact, under our experimental conditions, the C18 dimer fraction in the 1,3,5-TMB experiments was around 86.5%, which is much higher than the dimer fraction of 42.6%-56.5% re-calculated using the measured C9 and C18 signals by Tsiligiannis et al. (2019), 43.3%-52.4% modelled by Tsiligiannis et al. (2019), and 39% reported by Molteni et al. (2018)....",

and in Section S1 of the supplement that "PAM_chem_v8 is a model developed in conjunction with the PAM, which includes the chemistry of photolysis of oxygen, water vapor, and other trace gases by the primary wavelengths in mercury lamps (254 nm and 185 nm) (Lambe et al., 2017; Li et al., 2015; Peng et al., 2015). Simplified VOC and RO$_2$ chemistry are also included, but the first-generation stabilized products and the second-generation organic radical products do not react further in the model.

Fig S4 shows the modelled concentration profiles of different oxidants and 1,3,5-TMB with an irradiance of $1.64 \times 10^{15}$ ph cm$^{-2}$ s by 254 nm lamps. The initial concentrations of $O_3$ ([$O_3$]) (1.2 ppm) and 1,3,5-TMB ([1,3,5-TMB]) were measured before turning on the 254 nm lamps. [$O_3$] and [1,3,5-TMB] at an 80 s residence time were also measured. The modelled [$O_3$] is 20% lower than the measured value whereas modelled [1,3,5-TMB] is very close (4% higher) to the measured one, which shows the reliability of this model.

[Figure]

**Figure S4.** Concentration profiles of different oxidants and 1,3,5-TMB outputted by PAM_chem_v8 under the settings of Exp. #2. Initial [$O_3$] and [1,3,5-TMB] are 1.2 ppm and 118 ppb, respectively, which were used as input of the model. The measured [$O_3$] and [1,3,5-TMB] at the exit of OFR are shown by a triangle and a diamond, respectively. Input of irradiance of 254 nm lamps, $I_{254}$, is $1.64 \times 10^{15}$ ph cm$^{-2}$ s, which was measured with a photodiode in the OFR. The vertical purple line represents a residence time of 80 s"

**References:**

Berndt, T., Scholz, W., Mentler, B., Fischer, L., Herrmann, H., Kulmala, M. and Hansel, A.: Accretion Product Formation from Self- and Cross-Reactions of RO 2 Radicals in the Atmosphere, Angew. Chemie Int. Ed., 57(14), 3820–3824, doi:10.1002/anie.201710989, 2018.

Ehn, M., Thornton, J. A., Kleist, E., Sipilä, M., Junninen, H., Pullinen, I., Springer, M., Rubach, F., Tillmann, R., Lee, B., Lopez-Hilfiker, F., Andres, S., Acir, I. H., Rissanen, M., Jokinen, T., Schobesberger, S., Kangasluoma, J., Kontkanen, J., Nieminen, T., Kurtén, T., Nielsen, L. B., Jørgensen, S., Kjaergaard, H. G., Canagaratna, M., Maso, M. D., Berndt, T., Petäjä, T., Wahner, A., Kerminen, V. M., Kulmala, M., Worsnop, D. R., Wildt, J. and Mentel, T. F.: A large source of low-volatility secondary organic aerosol, Nature, 506(7489), 476–479, doi:10.1038/nature13032, 2014.

Garmash, O., Rissanen, M. P., Pullinen, I., Schmitt, S., Kausiala, O., Tillmann, R., Zhao, D., Percival, C., Bannan, T. J., Priestley, M., Hallquist, Å. M., Kleist, E., Kiendler-Scharr, A., Hallquist, M., Berndt, T., McFiggans, G., Wildt, J., Mentel, T. F. and Ehn, M.: Multi-generation OH oxidation as a source for highly oxygenated organic molecules from aromatics, Atmos. Chem. Phys., 20(1), 515–537, doi:10.5194/acp-20-515-2020, 2020.

de Gouw, J. and Warneke, C.: Measurements of volatile organic compounds in the earth's atmosphere using proton-transfer-reaction mass spectrometry, Mass Spectrom. Rev., 26(2), 223–257, doi:10.1002/mas.20119, 2007.

Gueneron, M., Erickson, M. H., Vanderschelden, G. S. and Jobson, B. T.: PTR-MS fragmentation patterns of gasoline hydrocarbons, Int. J. Mass Spectrom., 379, 97–109, doi:10.1016/j.ijms.2015.01.001, 2015.

Hyttinen, N., Kupiainen-Määttä, O., Rissanen, M. P., Muuronen, M., Ehn, M. and Kurtén, T.: Modeling the Charging of Highly Oxidized Cyclohexene Ozonolysis Products Using Nitrate-Based Chemical Ionization, J. Phys. Chem. A, 119(24), 6339–6345, doi:10.1021/acs.jpca.5b01818, 2015.

Karl, T., Striednig, M., Graus, M., Hammerle, A. and Wohlfahrt, G.: Urban flux measurements reveal a large pool of oxygenated volatile organic compound emissions, Proc. Natl. Acad. Sci. U. S. A., 115(6), 1186–1191, doi:10.1073/pnas.1714715115, 2018.

Lambe, A., Massoli, P., Zhang, X., Canagaratna, M., Nowak, J., Daube, C., Yan, C., Nie, W., Onasch, T., Jayne, J., Kolb, C., Davidovits, P., Worsnop, D. and Brune, W.: Controlled nitric oxide production via O(1D) + N2O reactions for use in oxidation flow reactor studies, Atmos. Meas. Tech., 10(6), 2283–2298, doi:10.5194/amt-10-2283-2017, 2017.

Li, R., Palm, B. B., Ortega, A. M., Hlywiak, J., Hu, W., Peng, Z., Day, D. A., Knote, C., Brune, W. H., De Gouw, J. A. and Jimenez, J. L.: Modeling the radical chemistry in an oxidation flow reactor: Radical formation and recycling, sensitivities, and the OH exposure estimation equation, J. Phys. Chem. A, 119(19), 4418–4432, doi:10.1021/jp509534k, 2015.

Molteni, U., Bianchi, F., Klein, F., Haddad, I. El, Frege, C., Rossi, M. J., Dommen, J. and Baltensperger, U.: Formation of highly oxygenated organic molecules from aromatic compounds, Atmos. Chem. Phys, 18, 1909–1921, doi:10.5194/acp-18-1909-2018, 2018.

Peng, Z. and Jimenez, J. L.: Radical chemistry in oxidation flow reactors for atmospheric chemistry research, Chem. Soc. Rev., 49(9), 2570–2616, doi:10.1039/c9cs00766k, 2020.

Peng, Z., Day, D. A., Stark, H., Li, R., Lee-Taylor, J., Palm, B. B., Brune, W. H. and Jimenez, J. L.: HO x radical chemistry in oxidation flow reactors with low-pressure mercury lamps systematically examined by modeling, Atmos. Meas. Tech, 8, 4863–4890, doi:10.5194/amt-8-4863-2015, 2015.

Peng, Z., Palm, B. B., Day, D. A., Talukdar, R. K., Hu, W., Lambe, A. T., Brune, W. H. and Jimenez, J. L.: Model Evaluation of New Techniques for Maintaining High-NO Conditions in Oxidation Flow Reactors for the Study of OH-Initiated Atmospheric Chemistry, ACS Earth Sp. Chem., 2(2), 72–86, doi:10.1021/acsearthspacechem.7b00070, 2018.

Španěl, P. and Smith, D.: On the features, successes and challenges of selected ion flow tube mass spectrometry, Eur. J. Mass Spectrom., doi:10.1255/ejms.1240, 2013.

Tani, A.: Fragmentation and reaction rate constants of terpenoids determined by proton transfer reaction-mass spectrometry, Environ. Control Biol., 51(1), 23–29, doi:10.2525/ecb.51.23, 2013.

Yuan, B., Koss, A. R., Warneke, C., Coggon, M., Sekimoto, K. and De Gouw, J. A.: Proton-Transfer-Reaction Mass Spectrometry: Applications in Atmospheric Sciences, Chem. Rev., 117(21), 13187–13229, doi:10.1021/acs.chemrev.7b00325, 2017.